



# Research on Hydrogeochemical Characteristics and Transformation Relationships
# between Surface Water and Groundwater in the Weihe River
*Jihong Qu* [1], *Shibao Lu*[2,*], *Zhipeng Gao* [3,1], *Wujin Li*[4, 1], *Zhiping Li*[1], *Furong Yu*[1]
(1 *School of Resources and Environment, North China University of Water Resources and Electric Power, Zhengzhou 450045, China*
2 *School of Public Administration, Zhejiang University of Finance and Economics, Hang Zhou 310018, China;*
3 *School of Water Resources and Environment, China University of Geosciences (Beijing), Beijing 100083, China*
4 *Guangdong Hydropower Planning and Design Institute, Guangzhou 510635, China* )
[*]*corresponding author:Lu5111284@aliyun.com*
**Abstract:** The transforming relationship between surface water and groundwater as well as their origins are the basis for studying the
transport of pollutants in river-groundwater systems. A typical section of the river was chosen to sample the surface water and
shallow groundwater. Then, a Piper trilinear diagram, Gibbs diagram, ratios of major ions, factor analysis, cluster analysis and other
methods were used to investigate the hydrogeochemical evolution of surface water and groundwater and determine the formation of
hydrogeochemical components in different water bodies. Based on the distribution characteristics of hydrogen and oxygen stable
isotopes $\delta D$ and $\delta^{18}O$ and discharge hydrograph separation methods, the relationship between surface water and groundwater in the
Weihe River was analyzed. The results indicated that the river water is a $SO_4 \cdot Cl$—Na type and that the groundwater
hydrogeochemical types are not the same. The dominant anions are $HCO_3^-$ in the upstream reaches and are $SO_4^{2-}$ and $Cl^-$ in
downstream reaches. Hydrogeochemical processes include evaporation and concentration, weathering of rocks, ion exchange, and
dissolution infiltration reactions. The $\delta D$ and $\delta^{18}O$ of surface water change little along the river and are more enriched than are those
of the groundwater. With the influences of precipitation, irrigation, river recharge and evaporation, the $\delta D$ and $\delta^{18}O$ of shallow
groundwater at different sections are not the same. There is a close relationship between the surface water and groundwater. Surface
water supplies the groundwater, which provides the hydrodynamic conditions for the entry of pollutants into the aquifer.
**Keywords:** hydrogeochemical characteristics; hydrogen and oxygen stable isotopes; surface water-groundwater system; cycle and
transformation

## 1. Introduction
The regularity of the water cycle and the conversion between surface water and groundwater is the basis for
the study of pollutant transport in river-groundwater systems (Lu et al., 2016a; Shang et al., 2017). Different water
bodies have different hydrogeochemical characteristics and isotopic signatures because of their different sources
of recharge, environments and circulation conditions. Hence, hydrogeochemical characteristics are the ideal
tracers for tracking water circulation processes. Analyses of hydrogeochemical and isotopic characteristics of
rivers and groundwater can effectively reveal the relationship in the transformation of river water to groundwater.
Descriptive statistics, graphic analysis and multivariate statistical analysis are used to analyze the
hydrogeochemical characteristics (Lu et al., 2015). Graphic methods include the Durov diagram, Stiff diagram,
Piper diagram and Gibbs diagram. Common multivariate statistical analysis methods include factor analysis
(Keesari et al., 2016), principal component analysis (Chattopadhyay and Singh, 2103) and cluster analysis (Zhang
et al., 2012). Stable isotopes of $\delta D$ and $\delta^{18}O$ are ideal tracers for tracking various hydrogeochemical processes
(Liu et al., 2014). Dogramaci et al. (2012) studied the hydrogeochemical and isotopic characteristics of the
Hamersley Basin in northwestern Australia and provided a theoretical basis for the sustainable development of
local water resource utilization. The descriptive statistical method, the Piper diagram and the main ion component
proportion coefficient and factor analysis method were used to study hydrogeochemical characteristics of





groundwater in the Sara Wusu aquifer system in the Ordos Basin (Yang et al., 2016). Fuzzy mathematics and multivariate statistical methods were used to study the quality characteristics of surface water and groundwater in the Songnen plain (Zhang et al., 2012). Zeng et al. (2013) studied the spatial distribution of hydrogeochemical and isotopic characteristics of different water bodies, including spring water, river water and lake water, in different parts of Tajikistan and discussed their origins and environmental significance. Although there are many studies related to the chemical and isotopic characteristics of groundwater and surface water, the relationship between surface water and groundwater transformation is still a prevalent topic in hydrology and water resource studies (Wang et al., 2016; Lu et al., 2016b), hydrogeochemistry, biogeochemistry, and ecohydrology. Hydrogen and oxygen stable isotopes ($\delta^{18}O$ and $\delta D$) and electrical conductivity (EC) were used to study the mutual relationship among precipitation, river water and groundwater in Taiwan Douliushan (Peng et al., 2014). Multivariate statistical analysis methods and isotope analysis methods were used to study the hydraulic linkage between surface water and ground water and their temporal and spatial variation in the Condamine River in Australia (Martinez et al., 2015). Hydrogen and oxygen isotopes were used to study the relationship of recharge and discharge between the various water bodies on the Portuguese island of Madeira, from which a hydrogeological conceptual model of Madeira Island was established (Prada et al., 2016). By analyzing the hydrogeochemical characteristics of surface water and groundwater in the Heihe River Basin, Nie et al. (2005) identified the transformation relationship between groundwater and surface water in the main stream of Heihe River. Hydrogen and oxygen isotopes and water chemistry were used to investigate the relationship between surface water and groundwater of the Second Songhua River, and the end element method was used to quantitatively calculate a conversion proportion between surface water and groundwater (Zhang et al, 2014).

The surface water of Weihe River is seriously polluted and has become a major pollution source for nearby shallow groundwater. This seriously affects the exploitation, utilization and protection of groundwater resources and endangers the ecological safety and the health of the residents (Lu et al., 2016c). The surface water and groundwater were sampled in several typical sections of the Weihe River Basin to study the conversion relationship between surface water and groundwater based on the hydrogeochemical and isotopic characteristics and to provide a basis for groundwater protection, restoration and management.

## 2. Materials and methods

### 2.1 Study area

The Weihe River Basin, which is 344.5 km long and has a basin area of 14970 km², is located in the northern part of the Henan Province, south of the North China Plain. It is the main tributary to the Zhangweinan Canal, which is a tributary to the Haihe River. The Weihe River Basin has a warm, temperate, continental monsoon climate. It is cold, with minor rain in the winter, hot and rainy in the summer, and the average annual precipitation in the basin is 608 mm (Zhu et al., 2006). The influence of river pollutants on groundwater is mainly banded and has a relatively small area of influence. A 26.67-km-long segment of the Weihe River between Xizhangzhuang village of Xiaohe Town and Dongwangqiao village of Liyang Town, considering the shallow groundwater along both sides of the river, was selected as the study area. This is an area of approximately 160 km² (shown in Fig. 1). The Weihe River Basin is closely related to groundwater, and the polluted river water of the Weihe River is a pollution source of groundwater on both sides of the river. The groundwater is mainly supplied by atmospheric precipitation, lateral seepage, piedmont lateral runoff and canal leakage, and drainage is dominated by artificial extraction and evaporation. Groundwater flows from the southwest to the northeast, which is generally consistent with the topography. The average hydraulic gradient is 1/3000.





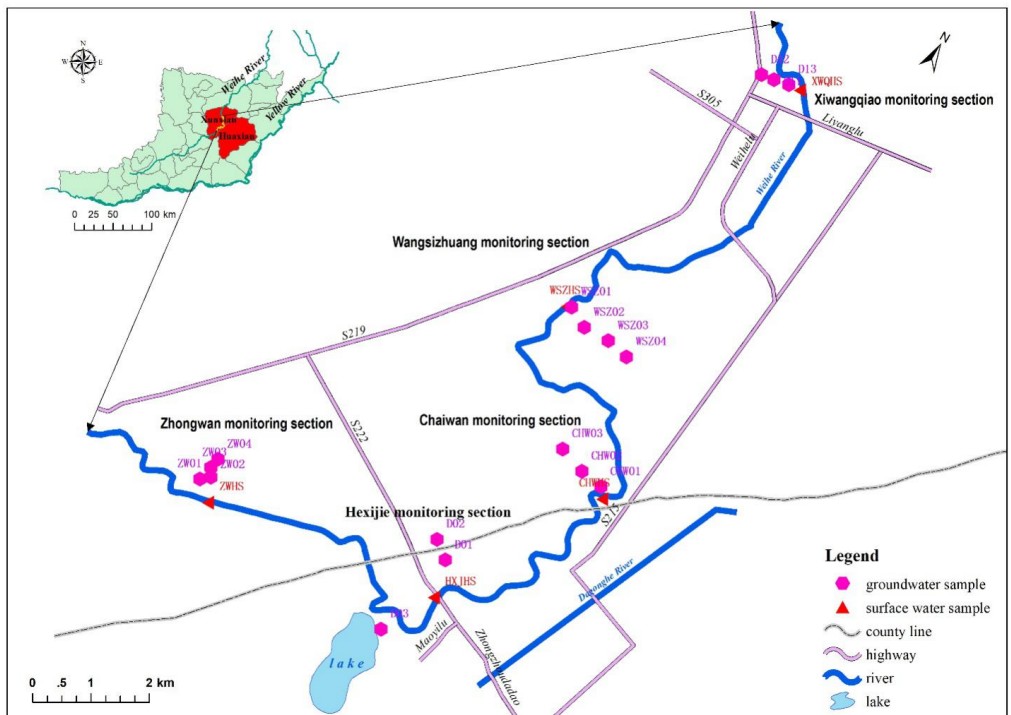

Fig. 1 The location of the study area and the distribution of sampled sections

## 2.2 Sample collection

According to the goals of the study, the surface water sampling sites were chosen parallel to the direction of river flow, and the groundwater sampling sites were chosen perpendicular to the river flow. There were 5 sections sampled from upstream to downstream between Xiaohe Town and Liyang Town along the Weihe River (i.e., the Zongwan sample section, Hexijie sample section, Chaiwan sample section, Wangsizhuang sample section and Xiwangqiao sample section). A total of 5 surface water samples and 17 groundwater samples were collected. Among them, ZWHS, HXJHS, CHWHS, WSZHS and XWQHS were surface water sampling sites, and the others were groundwater sampling sites. The samples were collected in May 2016. The sampling sites encompassed the band of influence of pollutants from the Weihe River on groundwater. The locations of the sampling sites and the types of water samples are shown in Figure 1.

Groundwater was sampled mainly from irrigation wells and drinking wells. Prior to sampling, wells were pumped for more than 20 min until the temperature, EC, and pH were stable. Surface water samples were collected from the river bank at a depth of more than 50 cm. Samples were collected in 500 ml polyethylene bottles, which were cleaned with sample water at least three times prior to sampling. When each sample was collected, no bubbles were left in the bottle, and the outer cap was sealed with sealant to prevent air exchange. Samples were brought back to the laboratory and stored in a refrigerated container at 0 to 4 ℃.

## 2.3 Stable hydrogen and oxygen isotopes and hydrogeochemical analysis

Hydrogen and oxygen isotopes were measured in the laboratory of groundwater science and engineering of the Ministry of Land and Resources of the Institute of Hydrogeology and Environmental Geology at the Chinese




Academy of Geological Sciences. Analyses were conducted using wavelength scanning-optical cavity ring down
spectroscopy. The ratio of hydrogen and oxygen isotopes ($\delta$) is expressed as the deviation relative to Vienna
VSMOW (Zhao, et al, 2015):
$$\delta(\text{‰}) = \frac{R_{sp} - R_{st}}{R_{st}} \times 1000 \qquad (1)$$

where $R_{sp}$ and $R_{st}$ refer to the ratio of D/H (or $^{18}O/^{16}O$) in samples and VSMOW, respectively. When $\delta D$ and
$\delta^{18}O$ are positive, the samples are enriched with D and $^{18}O$ compared to the VSMOW standard; when they are
negative, the two isotopes are diluted compared to the VSMOW standard (Zhang et al., 2006).
Analyses of water chemistry components were completed in the laboratory of hydrogeology at the North
China University of Water Resources and Electric Power. The analyses included $Cl^-$, $SO_4^{2-}$, $Na^+$, $K^+$, $NH_4^+$, $Mg^{2+}$,
$HCO_3^-$, $CO_3^{2-}$ and $Ca^{2+}$. Among these ions, $HCO_3^-$ and $CO_3^-$ were detected using acid-base indicator titration, $Ca^{2+}$
and $Mg^{2+}$ were detected using EDTA titration, and the other ions were detected using ion chromatography. pH,
TDS, RDO, conductivity, redox potential and other indicators were detected *in situ* with a PX.68-smarTROLL
MP hand-held multi-parameter water quality detector.

**2.4 Conversion ratio of surface water to ground water**


The stable hydrogen and oxygen isotope method can determine the sources of runoff, the division of river
runoff and the conversion of surface water and groundwater. The principle of division is based on the mass
conservation of isotopes (Song et al., 2007), in which the sum of two runoff components is equal to the flow of the
resultant runoff, and the sum of the tracer flow of the two runoff components is equal to the sum of the tracer of
synthetic runoff (Figure 2). The calculations are as follows:
$$Q_t = Q_u + Q_v \qquad (2)$$

$$Q_t \cdot C_t = Q_u \cdot C_u + Q_v \cdot C_v \qquad (3)$$

$$f = \frac{Q_v}{Q_t} = \frac{C_t - C_u}{C_v - C_u} \qquad (4)$$

where $Q$ is the flux, $C$ is the isotope component, $t$ and $u$ are surface water, and $v$ is groundwater. $f$ is the ratio
of river water to ground water and is calculated with $\delta D$ as a standard.

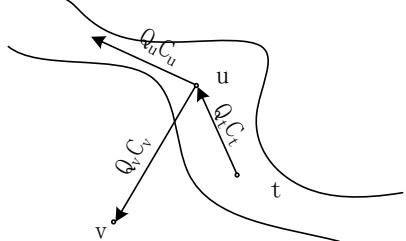


**Fig. 2 Principle diagram of the discharge hydrograph separation methods**





## 3. Results and discussion

### 3.1 Characteristics of main hydrogeochemical components

The water composition results are shown in Table 1. The groundwater pH in the study area was near neutral, ranging from 6.83 to 7.81. The TDS ranged from 564.66 to 1747.84 mg/L, and the TDS of all 17 groundwater samples exceeded the WHO drinking water standard of 500 mg/L. The relationship between the average concentrations of groundwater anions was $HCO_3^- > SO_4^{2-} > Cl^-$. The concentration of $HCO_3^-$ ranged from 461.75 mg/L to 735.15 mg/L, with an average concentration of 635.88 mg/L. The concentration of $SO_4^{2-}$ ranged from 116.11 mg/L to 833.33 mg/L, with an average concentration of 307.52 mg/L. The concentration of $Cl^-$ ranged from 102.17 mg/L to 640.13 mg/L, with an average concentration of 275.24 mg/L. The relationship of the average concentrations of the cations was $Na^+ > Ca^{2+} > Mg^{2+} > K^+$. $Na^+$ and $Ca^{2+}$ were dominant, and their concentrations ranged from 74.21 mg/L to 272.00 mg/L and 64 mg/L to 268.80 mg/L, respectively, with average values of 182.78 mg/L and 121.69 mg/L.

The pH of the Weihe River in the study area ranged from 8.03 to 8.22 and was therefore weakly alkaline. The TDS ranged from 1401.32 to 1518.71 mg/L, which is generally higher than that of groundwater. The relationships between the concentrations of anions and cations in surface water were $SO_4^{2-} > Cl^- > HCO_3^-$ and $Na^+ > Ca^{2+} > Mg^{2+} > K^+$. The concentration of $SO_4^{2-}$ in the river water ranged from 627.07 mg/L to 664.06 mg/L, with an average value of 647.12 mg/L. The concentration of $Cl^-$ ranged from 325.95 mg/L to 391.57 mg/L, with an average concentration of 365.89 mg/L. The concentrations of $Na^+$ and $Ca^{2+}$ ranged from 294.47 mg/L to 314.27 mg/L and 94.40 mg/L to 115.20 mg/L, respectively, and their mean values were 305.58 mg/L and 107.52 mg/L. As seen in Table 1, there is no significant change in the ion concentration between the upstream and downstream parts of the Weihe River.

According to WHO standard of drinking water, except for $K^+$ and pH, the other measured components of surface water and groundwater all exceeded the maximum acceptable values in the study area. As such, both surface water and groundwater along the Weihe River are not suitable drinking water sources.

**Table 1 Analytical results of water quality in the study area**

| Ion content/ (mg.L$^{-1}$) | | pH | TDS | Na$^+$ | K$^+$ | Mg$^{2+}$ | Ca$^{2+}$ | Cl$^-$ | SO$_4^{2-}$ | HCO$_3^-$ | TH |
|---|---|---|---|---|---|---|---|---|---|---|---|
| Groundwater (17) | Minimum | 6.73 | 564.66 | 74.21 | 6.20 | 57.76 | 64.00 | 102.17 | 116.11 | 461.75 | 396.83 |
| | Maximum | 7.81 | 1747.84 | 272.00 | 34.26 | 162.38 | 268.80 | 640.13 | 833.33 | 735.15 | 1072.91 |
| | Average | 7.30 | 1170.00 | 182.78 | 16.58 | 110.23 | 121.69 | 275.24 | 307.52 | 635.88 | 756.18 |
| Surface water (5) | Minimum | 8.03 | 1401.32 | 294.47 | 24.23 | 49.90 | 94.40 | 325.95 | 627.07 | 282.52 | 480.13 |
| | Maximum | 8.22 | 1518.71 | 314.27 | 28.35 | 59.54 | 115.20 | 391.57 | 664.06 | 385.80 | 495.84 |
| | Average | 8.11 | 1473.74 | 305.58 | 26.40 | 53.79 | 107.52 | 365.89 | 647.12 | 350.56 | 489.33 |
| WHO drinking water standards | | 6.5～8.5 | 500 | 200 | 100 | 30 | 75 | 200 | 200 | 200 | 100 |
| Over-standard rate of groundwater (%) | | 0 | 100 | 47 | 0 | 100 | 70 | 70 | 65 | 100 | 100 |
| Over-standard rate of surface water (%) | | 0 | 100 | 100 | 0 | 100 | 100 | 100 | 100 | 100 | 100 |

### 3.2 Hydrogeochemical characteristics

The Piper diagram is one of the most commonly used graphical methods for interpreting hydro-geological problems. According to the analytical results, the Piper diagram of the hydrogeochemical composition of all water samples in the study area is shown in Fig. 3. The results indicate that the chemical type of surface water in the study area is the $SO_4 \cdot Cl$—Na type, indicating that the surface water is uniform across the study area. From





upstream to downstream along the Weihe River, the water chemistry type of each groundwater section is as
follows: The Zong Wan section and Hexijie section are mainly HCO₃--Mg·Na types, the Chaiwan section is
mainly the SO₄·Cl—Mg·Na type, the Wangsizhuang section is mainly the HCO₃·SO₄·Cl--Mg·Na type, and the
Xiwangqiao section is mainly the HCO₃·Cl—Mg·Ca·Na type. Usually, the chemical types of groundwater, from
the recharge area to the discharge area, change in the following ways: $HCO_3^-$ —$SO_4^{2-}$ —$Cl^-$. Considering the
chemical types of groundwater of each section, $HCO_3^-$ is dominant in groundwater on both sides of river in the
upstream section, whereas $SO_4^{2-}$ and $Cl^-$ are dominant in the middle and lower reaches. The type of water
chemistry can indirectly verify the groundwater flow on both sides of river.

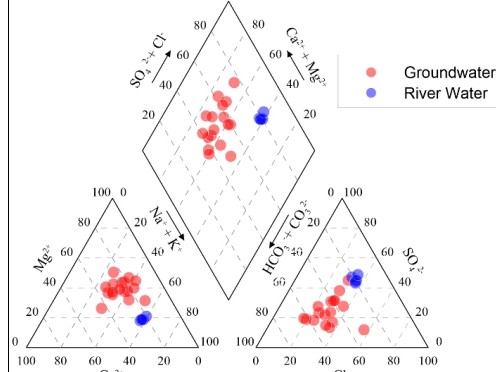

**Fig. 3 Piper diagram of water chemistry for surface and groundwater in the study area**
**3.3 Analysis of formation function of water chemical composition**
**3.3.1 Analysis of formation function based on the Gibbs diagram**
The Gibbs diagram can clearly indicate whether the chemical components of river and groundwater are the
precipitation dominance type, rock dominance type or evaporation crystallization dominance type. It is an
important way to qualitatively determine the effects of regional rocks, atmospheric precipitation, and evaporation
concentration on river water components (Wang et al., 2010). Generally, samples with low TDS and high
$Na^+/(Na^++Ca^{2+})$ or $Cl^-/(Cl^-+HCO_3^-)$ ratios (close to 1) are mainly distributed in the lower-right corner, indicating
precipitation dominance. Samples with slightly high TDS and $Na^+/(Na^++Ca^{2+})$ or $Cl^-/(Cl^-+HCO_3^-)$ ratios of
approximately 0.5 or less than 0.5 are mainly distributed in the middle zone, indicating rock dominance. Samples
with very high TDS and large $Na^+/(Na^++Ca^{2+})$ or $Cl^-/(Cl^-+HCO_3^-)$ ratios are mainly distributed in the upper-right
corner, indicating evaporation crystallization dominance type, reflecting the influence of evaporation in arid areas
(Sun et al., 2014).
The ion concentrations of the 5 river groundwater samples and 17 groundwater samples from the study area
are shown on a Gibbs diagram in Fig. 4. It is apparent that the surface water in the study area is located the upper-
right corner of the diagram with a $Na^+/(Na^++Ca^{2+})$ or $Cl^-/(Cl^-+HCO_3^-)$ ratio greater than 0.5 and with a high
content of TDS, indicating that surface water has an evaporation crystallization dominance origin. The
groundwater samples differ in the two figures but are mainly distributed in the evaporation crystallization
dominance region, slightly toward the rock dominance region, indicating that the chemical composition of water
is controlled by evaporation crystallization and rock weathering.





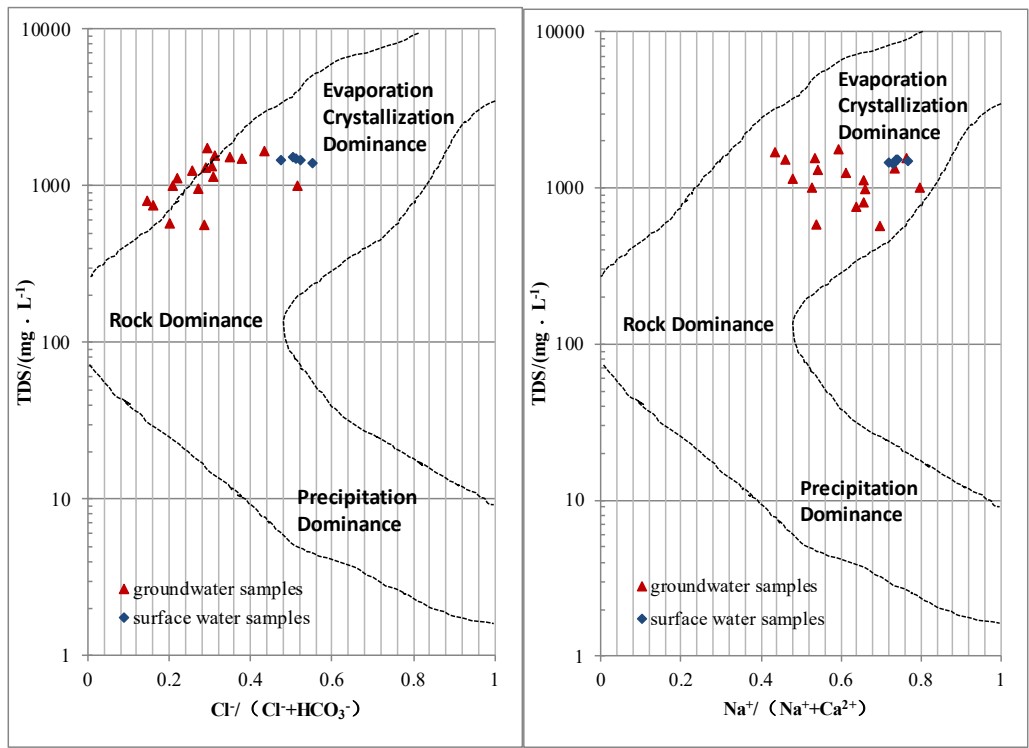


**Fig. 4 Gibbs plots of the surface and groundwater chemistry in the study area**
**3.3.3 Analysis of formation function based on multivariate statistics**
To further analyze the hydrogeochemical formation functions, factor analysis and R cluster analysis were
performed on water samples using TDS, $Cl^-$, $SO_4^{2-}$, $HCO_3^-$, $Na^+$, $K^+$, $Ca^{2+}$, and $Mg^{2+}$ as the original parameters.
Seventeen groups of groundwater samples and 5 groups of surface water samples were calculated. After the
calculation, the KMO value of the groundwater sample was 0.596. According to the KMO test standard, when
$0.5 < KMO < 0.6$, the original variable is barely suitable for factor analysis. According to the calculation results
(Table 2), the formation function of the groundwater chemical composition can be summarized in 3 factors, and
the cumulative contribution rate of the 3 factors is 75%. The $Ca^{2+}$, $Cl^-$, $SO_4^{2-}$ and TDS of factor 1 have higher
positive load and the coefficient of $Ca^{2+}$ and $SO_4^{2-}$ is large, indicating that there may be weathering of calcium
feldspar, dissolution of gypsum, or oxidation of pyrite. A large coefficient of $Cl^-$ indicates that there may be
leaching of halite. The $Na^+$, $Mg^{2+}$ and $HCO_3^-$ of factor 2 have higher positive loads, which indicates possible
weathering of carbonate or silicate. The alternating adsorption of cations between $Na^+$ and $Ca^{2+}$ causes the content
of $Na^+$ to increase. The $K^+$ content of factor 3 is large, which indicates possible weathering of feldspar. Surface
water samples can be summarized in 2 factors, and the contribution rate of the 2 factors is 91%. The $K^+$, $HCO_3^-$
and TDS of factor 1 have higher positive loads, indicating the possible weathering of carbonate and feldspar. The
$Mg^{2+}$, $Na^+$ and $Cl^-$ of factor 2 have a higher positive load, which indicates the possible dissolution of halite and the
alternate adsorption of cations between $Na^+$ and $Ca^{2+}$, which causes the content of $Na^+$ to increase.
Cluster analysis can be simplified as the identification of the relationship between large-scale samples. R
cluster analysis is used to classify variables, and Q cluster analysis is used to classify samples (Sun and Gui,




2013). The results of the R cluster analysis are shown in Figure 5. The groundwater components can be divided
into 3 groups, which is consistent with the results of the factor analysis. The first group includes $Ca^{2+}$, $Cl^-$, $SO_4^{2-}$
and TDS. The second group includes $Na^+$, $Mg^{2+}$ and $HCO_3^-$. Finally, the third group includes $K^+$. Surface water
ions can be divided into 2 groups. The first group includes TDS, $Cl^-$, $HCO_3^-$, $Na^+$, $K^+$ and $Mg^{2+}$, and the second
group includes $Ca^{2+}$ and $SO_4^{2-}$. This indicates the presence of gypsum dissolution, which differs from the results
of factor analysis.
In summary, a variety of complex hydrogeochemical processes may have occurred in the study area, such as
concentration through evaporation, rock weathering, cation alternate adsorption, oxidation and dissolution.
**Table 2 Factor analysis composition coefficient of ground and surface water**

| Parameter variable | Groundwater | | | Surface water | |
|---|---|---|---|---|---|
| | Factor 1 | Factor 2 | Factor 3 | Factor 1 | Factor 2 |
| $Ca^{2+}$ | 0.889 | -0.083 | 0.339 | 0.206 | -0.970 |
| $Na^+$ | 0.261 | 0.799 | -0.041 | 0.650 | 0.738 |
| $K^+$ | 0.061 | 0.081 | 0.916 | 0.783 | 0.540 |
| $Mg^{2+}$ | 0.298 | 0.702 | 0.265 | 0.106 | 0.986 |
| $Cl^-$ | 0.735 | 0.259 | 0.323 | 0.479 | 0.817 |
| $SO_4^{2-}$ | 0.760 | 0.299 | -0.046 | -0.883 | 0.351 |
| $HCO_3^-$ | -0.004 | 0.915 | 0.010 | 0.812 | 0.151 |
| TDS | 0.762 | 0.197 | -0.208 | 0.972 | 0.200 |
| Characteristic value | 3.472 | 1.492 | 1.037 | 4.856 | 2.446 |
| Contribution rate% | 43.394 | 18.649 | 12.965 | 60.704 | 30.578 |
| Cumulative contribution rate | 43.394 | 62.043 | 75.008 | 60.704 | 91.282 |

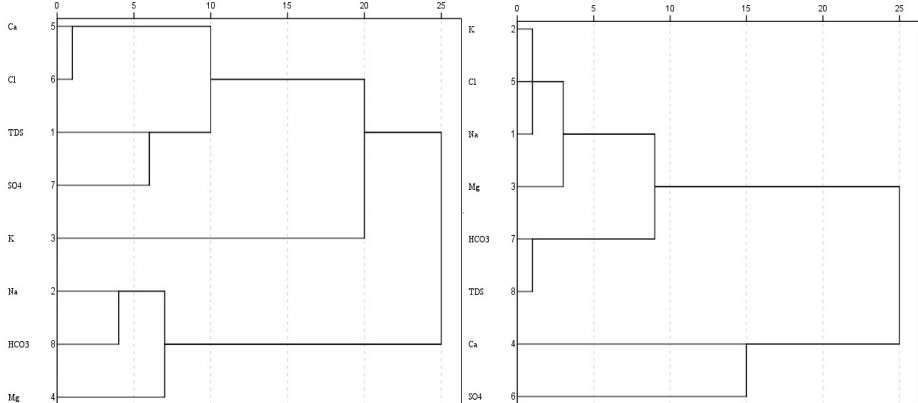

**Fig. 5 R type cluster analysis**

**3.4 Isotopic characteristics and transformation relationships of surface water and groundwater**
3.4.1 Isotopic variation characteristics
δD and $δ^{18}O$ of surface water and groundwater and the *d* value of deuterium excess in the study area are
listed in Table 3, in which $d=δD-8δ^{18}O$.
As seen in Table 3, δD and $δ^{18}O$ of surface water are more enriched than in groundwater. The variation in δD
and $δ^{18}O$ for water from the Weihe River is small. $δ^{18}O$ ranges from -7.8‰ to -7.6‰ with an average value of -





7.7‰, and δD ranges from -59‰ to -57‰ with an average value of -58‰. The range of δ18O for shallow
groundwater is from -9.4‰ to -7.7‰, with an average value of -8.55‰; the range of δD is from -59‰ to -69‰,
with an average value of -63.3‰. The *d* value of deuterium excess of water is positive and less than 10 of the
atmospheric precipitation intercept. That of surface water is less than that of shallow groundwater, indicating that
the recharge sources of surface water and groundwater are subject to evaporation effects but that shallow
groundwater is less influenced by evaporation effects.
Generally, the isotopic characteristics of river water bodies increase from upstream to downstream because
of the isotopic fractionation that is caused by the evaporation of water. The closer to the lower reaches of the river,
the greater the fractionation effect is on the isotopes (Liu et al., 2014). Figure 6 shows the variation in δD and
δ18O in river water. It is apparent that δD and δ18O become more enriched as the river flows downstream, in which
δ18O declines in HXJHS, probably because there is a lake in the vicinity of the upstream reaches and river water
supplies the lake.
For groundwater, the values of δD and δ18O in the Zongwan section, Hexijie section and Xiwangqiao section
become more depleted as the distance between sampling points and Weihe River increases. The closer the sample
location is to the river, the closer the δD and δ18O values are to the surface water, indicating that the influence of
surface water on groundwater decreases with increasing distance. In contrast, the influence of precipitation and
irrigation infiltration recharge on groundwater is enhanced. The values of δD and δ18O for the Chaiwan section
and the Wangsizhuang section become enriched as the distance between the sampling points and the Weihe River
increases. This is likely because the farmland in the Chaiwan section and the Wangsizhuang section is mainly
irrigated using Weihe River water, and the infiltration of irrigation water causes the enrichment of hydrogen and
oxygen isotopes in the groundwater. The hydrogen and oxygen isotope characteristics are more similar to those of
the Weihe River. The CHW02 and CHW03 sampling points in the Chaiwan section are located in an area affected
by river irrigation, and CHW01 is a household well. As such, the hydrogen and oxygen isotope values are
CHW03≥CHW02>CHW01. Similarly, WSZ02, WSZ03 and WSZ04 in the Wangsizhuang section are located in
an area affected by river irrigation, and WSZ01 is a household well. Thus, WSZ04≥WSZ03≥WSZ02> WSZ01.

**Table 3 δD, δ18Oand *d* values of water samples in study area**

| Water sample type | No. | δ D/‰ | δ18O/‰ | *d*/‰ | *f*/% |
|---|---|---|---|---|---|
| Surface water | ZWHS | -59 | -7.7 | 2.6 | |
| | HXJHS | -58 | -7.8 | 4.4 | |
| | CHWHS | -58 | -7.7 | 3.6 | |
| | WSZHS | -58 | -7.7 | 3.6 | |
| | XWQHS | -57 | -7.6 | 3.8 | |
| | average | -58 | -7.7 | 3.6 | |
| Shallow groundwater | ZW01 | -62 | -8.6 | 6.8 | 33.3 |
| | ZW02 | -63 | -8.6 | 5.8 | 25 |
| | ZW03 | -63 | -8.6 | 5.8 | 25 |
| | ZW04 | -69 | -9.4 | 6.2 | 10 |
| | D01 | -65 | -8.7 | 4.6 | 14.3 |
| | D02 | -66 | -8.9 | 5.2 | 12.5 |
| | D03 | -60 | -8 | 4 | 50 |
| | CHW01 | -63 | -8.6 | 5.8 | 20 |
| | CHW02 | -62 | -8.4 | 5.2 | 25 |
| | CHW03 | -62 | -8.3 | 4.4 | 25 |




| | | | | |
|---|---|---|---|---|
| WSZ01 | -68 | -9.2 | 5.6 | 10 |
| WSZ02 | -63 | -8.6 | 5.8 | 20 |
| WSZ03 | -63 | -8.5 | 5 | 20 |
| WSZ04 | -63 | -8.5 | 5 | 20 |
| D11 | -59 | -7.7 | 2.6 | 50 |
| D12 | -62 | -8.4 | 5.2 | 20 |
| D13 | -63 | -8.4 | 4.2 | 16.7 |
| Average | -63.3 | -8.55 | 5.1 | |


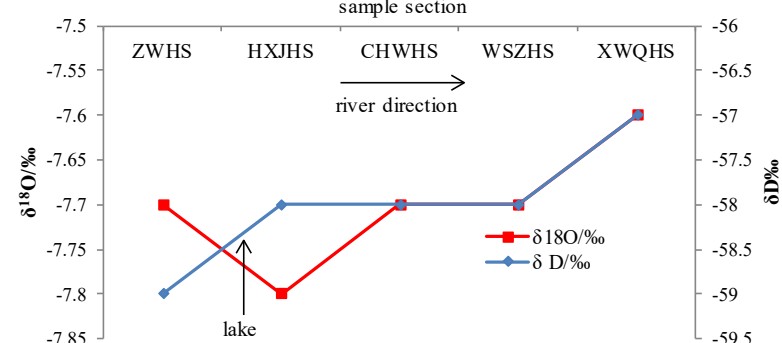
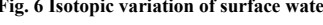


**Fig. 6 Isotopic variation of surface water**

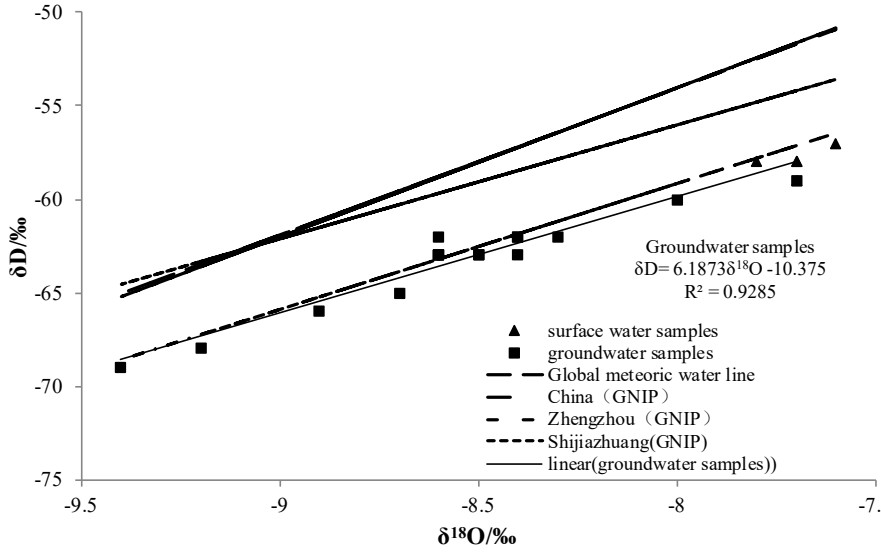


**Fig. 7 Relationship between δD and δ¹⁸O of surface water and groundwater**

According to the 27 GNIPs set up in China by the International Atomic Energy Association (IAEA), the

monitoring sites that are closest to the study area are at Shijiazhuang and Zhengzhou. The meteoric water line
$\delta D=6.75 \ \delta^{18}O-5.12$ in Zhengzhou is close to the characteristic line for hydrogen and oxygen isotopes of samples
in the study area (Figure 7), so the meteoric water line in Zhengzhou is assumed to be the local meteoric water



line (LMWL). When the compositions of δD and δ¹⁸O of water samples are compared to the meteoric water line,
the source of the local river water and shallow groundwater and their mutual transformation relationship can be
distinguished. From drawing the trend line between the underground water sample points, the relationship
between δD and δ¹⁸O is δD=6.1873δ¹⁸O-10.375, and the correlation coefficient is 0.9285. From drawing the trend
line between the groundwater and surface water sample points, the relationship between δD and δ¹⁸O is
δD=6.19328δ¹⁸O-10.321, and the correlation coefficient is 0.9585. The two trend lines are basically the same, and
the related coefficient is very high. The surface water sample points are located in the direction of the
groundwater trend line that extends to the right. δD and δ¹⁸O are relatively enriched, indicating that the sources of
surface water and groundwater are the same and that there is a hydraulic connection. The hydraulic connection
between the two is a single-line infiltration of the surface river water into groundwater. The trend line is close to
the local meteoric water line (LMWL) and the slope is small, indicating that surface water and groundwater are
recharged from meteoric water but are also subject to the evaporation, resulting in the enrichment of hydrogen and
oxygen isotopes.
3.4.2 Estimation of recharge capacity of river water to groundwater

According to the stable isotope signatures of the water samples, the calculated results of ratio *f* of
groundwater recharge to river water are shown in Table 4. The ratio *f* of surface water infiltration to recharge
groundwater in each observation section has a different, but regular, pattern, and the results are shown in Figure 8.
In the Zongwan section (Figure 8a), as the distance between groundwater sampling sites and the river
increased (ZW01 toward ZW04), the ratio of surface water infiltration to groundwater recharge (*f*) decreased from
33.3% to 10%, indicating that the river water recharges groundwater in this section and the direction of
groundwater flow is from ZW01 toward ZW04. The infiltration rates at D01 and D02 in the Hexijie section
(Figure 8b) are 14.3% and 12.5%, respectively, with a decreasing trend, indicating that there is a small amount of
river water recharging groundwater in that section, with a direction of groundwater flow from D01 toward D02.
The ratio of surface water infiltration to groundwater at D03 is as high as 50%, indicating that the river mainly
recharges the artificial lake that exists near D03 in the Hexijie section. The ratio of surface water infiltration to
groundwater in the Chaiwan section (Figure 8C) increases from 20% to 25% as the distance increases between
groundwater sampling sites and the river, whereas it increases from 10% to 20% in the Wangsizhuang section
(Figure 8D). This may be associated with the unique river trend of the two sections. The Chaiwan section and the
Wangsizhuang section are located near the right corner of the river, where the influence of the river water on
groundwater is complicated, but the river is the main supplier of groundwater. The groundwater flow line is the
closed space where water is not exchanged with the outside world. Some input values remain constant along the
entire streamline, such as δD and δ¹⁸O. Therefore, it is possible to interpret that WSZ01, WSZ02, and WSZ03 are
on the same streamline. At the same time, because farmland is primarily irrigated by water from the Weihe River,
irrigation water infiltrates the soil to recharge groundwater, resulting in the enrichment of hydrogen and oxygen
isotopes. For the Xiwangqiao section (Figure 8e), the ratio of river water infiltration to groundwater at D11 is
close to 50%, whereas it is 20% and 16.7% at D12 and D13, respectively. This is primarily because D11 is located
in the convexity of the river, where it is significantly eroded with a large amount of infiltration.


(a) Zongwan section

(b) Hexijie section

(c) Chaiwan section

(d) Wangsizhuang section

(e) Xiwangqiao section

**Fig. 8 Relationships between surface water and groundwater**
## 4. Conclusions
The surface water components of Weihe River display no significant spatial variation, but the ion
concentrations of groundwater samples from 5 sections are different. The cation concentrations of surface water
and groundwater are consistent, with $Na^+ > Ca^{2+} > Mg^{2+} > K^+$. The relative concentrations of anions in groundwater
are $HCO_3^- > SO_4^{2-} > Cl^- > NO_3^-$, whereas the relative concentrations of anions in the surface water are $SO_4^{2-} > Cl^- >$
$HCO_3^- > NO_3^-$. The surface water in all sections of the Weihe River is the $SO_4 \cdot Cl$—Na type, whereas the
hydrogeochemical types of groundwater are different. $HCO_3^-$ dominates in the groundwater in the upper reaches





of the river, and $SO_4^{2-}$ and $Cl^-$ dominate in the middle and lower reaches.
By using a Gibbs diagram, factor analysis and cluster analysis, we established that the geochemical processes
of the Weihe River Basin include concentration by evaporation, rock weathering, cation alternate adsorption and
dissolution. Because surface water is an open system, the source of ions in a water body is greatly influenced by
human activity and atmospheric precipitation, whereas the factors contributing to the formation of water
chemistry are more complex.
The isotope results show that $\delta D$ and $\delta^{18}O$ of the surface water in the Weihe River varies little and is more
enriched than the groundwater is. The shallow groundwater at different sections is affected by rainfall, irrigation,
river recharge and evaporation, resulting in different $\delta D$ and $\delta^{18}O$ values. By analyzing hydrogen and oxygen
isotopic characteristics of surface water and groundwater in different sections and using the segmentation of flow
duration curve, it was established that surface water recharges groundwater at 5 sections along the Weihe River,
and each section has unique recharge intensity and relationship due to its unique hydraulic environment.
Due to the lack of local isotope monitoring data for meteoric water, the Zhengzhou meteoric water line was
used to analyze the isotopic characteristics of surface water and groundwater. The existing household wells were
used as groundwater sampling points. Because they are affected by towns and villages surrounding the Weihe
River, groundwater sampling points cannot be fully symmetric and isometric relative to the Weihe River. As such,
the research results need to be improved by monitoring more complete data in future research.

## Acknowledgments

The study was financially supported by Non-Profit Industry Specific Research Projects of Ministry of Water
Resources, China, Grant NO: 201401041and 201501008, the Open Research Fund of State Key Laboratory of
Simulation and Regulation of Water Cycle in River Basin (China Institute of Water Resources and Hydropower
Research), Grant NO: IWHR-SKL-201208, and Science and Technology Research Key Project of the Education
Department of Henan Province, Grant NO: 14A170006.

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
