# Peer review of "Research on Hydrogeochemical Characteristics and Transformation Relationships"

_Hydrology and Earth System Sciences, 2017_

## Referee Comment (RC1) · X. Bai (Referee) · 15 Dec 2017

The work of the manuscript is of sufficient novelty, quality, and potential significance to the publication in the Journal of HESS. I believe the topic of this paper will be of great interest to the readers, and the concepts discussed in this paper are much needed in water resources community. In addition, this paper is well-written. Therefore, the publication is recommended. However, there are some points need to be elaborated before it can be accepted for publication. My comments are provided as follows: 1. The introduction should be rewritten, and a brief introduction of the work whether has

been done or not should be written thus highlighting the necessity and importance of this work 2. References are mostly Chinese literatures, it is recommended to add some English literatures. 3. It is suggested to modify the references format. 4. It is suggested to add the study area's location on the map of China in figure 1. 5. TH in Table 1 is not described in the text, which can be deleted. 6. The translation of name in figure 4 should be confirmed, which are Evaporation-crystallization dominance, Rock weathering dominance, Atmospheric precipitation dominance.

Please also note the supplement to this comment:
https://www.hydrol-earth-syst-sci-discuss.net/hess-2017-654/hess-2017-654-RC1-supplement.pdf

---

## Referee Comment (RC2) · Anonymous Referee #2 · 26 Dec 2017

This manuscript by Qu describes the groundwater hydrochemical and isotopic charateristics and relationships between surface water and groundwater in the Weihe River.There is no novel aspect on the methodology, such as Piper plot, Gibbs diagram, and ionic ratios, statistical analysis. The authors did not combined the analysis of hydrogeologic conditions with a database of chemical and isotopic data to illustrate the relationship between surface water and groundwater bodies.The whole paper lacks new understandings of interaction between surface- and ground-water in general. The manuscript is not organized well for a publication in a top international Journal. Additionally, it needs to convey some new understanding that ideally is applicable to other study areas. In the conclusions, the authors need to explain the relevance to research elsewhere. I consider that it does not constitute a valuable scientific contribution.

Abstract shows some results with lack of valuable conclusions. What's the evidence for "There is a close relationship between the surface water and groundwater." The authors should show some detailed data for the stable isotopic compostions in different water types in the ABSTRACT.

The INTRODUCTION does not provide good background to the study and places it in an international context. It is overelaborate without giving a straightforward idea of what the paper would like to present. The authors did not point out the specific aims of this study. What is not clear is what new general information you hope to provide. What new and scientific contribution wil come out of this paper?

Study Area: What's the main environmental issue associated with water resources in this study area? Please show the full statement for 'EC,TDS, RDO, WHO' when they first occur in the paper.

Materials and methods: Line 119:'The principle of division' For this method, does it have assumed condition? If use chloride or other tracers, how about the results? Line 127:'...is calculated with $\delta$D as a standard.', please explain it.

Results and Discussion: Section 3 is too long and tries to describe too many things. The thrust of this paper is to identify the interaction between surface water and groundwater. You should try to keep this as the main focus of the paper. Apart from 3.4.2, most of the contents in the section 3 are results. Line 259: for the data source from the 27 GNIPs set up in China, please add the specific references.

Conclusions: This section just summarises the main findings of the project. In this section explain in more detail how your project helps us to understand processes in these environments more broadly; the paper will have more impact if researchers from

[Figure]

elsewhere in the world can see relevance to their studies and a paper in a major international journal such as HESS needs to have broad appeal.

Figure 1, please show the coordinates of the map. Could you add the groundwater level contours on Fig. 1? Figure 5, is it for showing the results on groundwater samples? not clear. Please be prudent to use the cluster analysis and avoid the false correlation, especially for the groundwater samples. Figure 6,add 'a'and 'b' for the left and right diagrams, respectively. Add the corresponding figure captions. 'Isotopic variation of surface water' There is only one data for one reach of the river. What's the variation? Figure 7, the types of lines are not clear.

---

## Referee Comment (RC3) · Anonymous Referee #3 · 27 Dec 2017

The manuscript on "Research on Hydrogeochemical Characteristics and Transformation Relationships between Surface Water and Groundwater in the Weihe River" by Jihong Qu, et al. is trying to use graphic analysis and multivariate statistical analysis and stable isotopes to analyze the hydrochemical characteristics and the relationships between the river and groundwater in Weihe River. The purpose and the meaning of this study is not well stressed in the ABSTRACT. In addition, the authors need to put the results into a broader context. The data they used are not fully discussed combining the geological and hydrogeological settings of the study area.

In my opinion, the pollution in this study area is an aspect that we should pay attention to. But it didn't give any information about the pollutants. Will the pollutants change the hydrochemical contents of the waters?

For the STUDY AREA part, the Figure 1 of the sampling sites is not corresponding to the description, and it is not complimentary to enlighten the readers of the study area. For example, the places Xizhangzhuang village of Xiaohe Town and Dongwangqiao village of Liyang Town of the Line 76 are not showing in the figure. It would be helpful, if the contour of water table could be shown in the Figure 1.

In the DISCUSSION part, the interpretation of presented hydrochemical ions data could be supported with a more detailed description of the hydrological setting and lithology of the aquifer(s). All the discussion of the hydrochemical contents could go deeper if the geological settings were considered.

Generally, the manuscript is carelessly prepared. Text are readable, however, the abbreviations are not explained, and the figures are not well organized. So it is hard to read this manuscript clearly.

In details, what is the reproductivity of the hydrogen and oxygen isotopes? Please point out all Chinese references (in Chinese) for the international readers that do not understand Chinese language. It is hard to tell the difference of the lines in the Figure 7.

---

## Author Comment (AC1) · 24 Feb 2018

Q1: The introduction should be rewritten, and a brief introduction of the work whether has been done or not should be written thus highlighting the necessity and importance of this work.

A1: The introduction part has been rewritten, and key sentences have been added to address the importance and specific objectives of this project, which could be referred to lines 63-70.

[Figure]

Q2: References are mostly Chinese literatures, it is recommended to add some English literatures.

A2: Although most references in this paper are written by Chinese, the majority are published in top journals using English. Only two of them are in Chinese and have been labeled in the reference section. Also, we added several related literatures written by other countries' scholars to give a broader context of this topic.

Q3: It is suggested to modify the references format.

A3: The format of reference has been modified.

Q4: It is suggested to add the study area's location on the map of China in figure 1.

A4: We have demonstrated the location of Weihe river in the text. What we showed in Fig. 1 is the specific location of the study area in Weihe River basin, and the distribution of sampled sections, which are considered as more essential information.

Q5: TH in Table 1 is not described in the text, which can be deleted.

A5: Revised. The column of TH has been deleted.

Q6: The translation of name in figure 4 should be confirmed, which are Evaporation-crystallization dominance, Rock weathering dominance, Atmospheric precipitation dominance.

A6: The captain of Fig 4 has been revised.

---

## Author Comment (AC2) · 24 Feb 2018

Q1: Abstract shows some results with lack of valuable conclusions. What's the evidence for "There is a close relationship between the surface water and groundwater." The authors should show some detailed data for the stable isotopic compostions in different water types in the ABSTRACT.

A1: Thanks for your comments. We deleted the sentence of "There is a close relationship between the surface water and groundwater" and added "It could be established

that surface water recharges groundwater at 5 sections along the Weihe River, and each section has unique recharge intensity and relationship due to its specific hydraulic environment" of line 21-22.

Q2: The INTRODUCTION does not provide good background to the study and places it in an international context. It is overelaborate without giving a straightforward idea of what the paper would like to present. The authors did not point out the specific aims of this study. What is not clear is what new general information you hope to provide. What new and scientific contribution wil come out of this paper?

A2: Thanks for your comments. We have rewritten the introduction and added the specific aims at the end of this section. 'This paper has three main objectives as follows: (1) To investigate the hydrogeochemical components formation of surface water and groundwater through the samplings in several typical sections of the Weihe River Basin; (2) To determine the recharge relationship between surface water and groundwater based on the hydrogeochemical and isotopic characteristics; (3) To provide a series of methods for hydrogeochemical analysis including Piper trilinear diagram, Gibbs diagram, factor analysis and cluster analysis.'

Q3: Study Area: What's the main environmental issue associated with water resources in this study area? Please show the full statement for 'EC,TDS, RDO, WHO' when they first occur in the paper.

A3: Thanks for your comments. (1) There are two primary reasons why we choose this study area. Firstly, the shallow groundwater along both sides of the water could provide sufficient groundwater samplings for research. Secondly, the influence of river pollutants on groundwater is mainly banded and has a relatively small area of influence and the samplings would cover the influence of pollutants from the Weihe River on groundwater. (2) All the abbreviation statements are added when they first occur in the paper which are listed as follows: EC: electrical conductivity (lines 50 and 97) TDS: total dissolved solids (line 116) RDO: Rugged Dissolved Oxygen (line 116) WHO:

World Health Organization (line 135)

Q4: Materials and methods: : Line 119:'The principle of division' For this method, does it have assumed condition? If use chloride or other tracers, how about the results? Line 127:'...is calculated with D as a standard.', please explain it

A4: Thanks for your comments. 'The principle of division' is mainly referred to ' Song Xian-Fang, Liu Xiang-Chao, Xia Jun, Yu Jing-Jie, Tang Chang-Yuan. A study of interaction between surface water and groundwater using environmental isotopes in Huaisha River basin[J]. Science in China (Series D), 2007, 37(1):102-110.' It doesn't belong to assumed condition. And it needs further investigation If chloride or other tracers are used.

Q5: Results and Discussion: Section 3 is too long and tries to describe too many things. The thrust of this paper is to identify the interaction between surface water and groundwater. You should try to keep this as the main focus of the paper. Apart from 3.4.2, most of the contents in the section 3 are results. Line 259: for the data source from the 27 GNIPs set up in China, please add the specific references.

A5: Thanks for your comments. (1) We have collected 5 surface water and 17 groundwater samplings, then conducted hydrogeochemical tests and diagram analysis upon them. The results are shown in the 'Results and discussion' section, separated by different parts. Each subsection includes results and the according discussion, together with indications and suggestions through the reliable data from samplings in this study area. The results of both surface water and groundwater are the basis of relationship establishment, so we would like keeping the descriptions combined with discussion. (2) We added two references related to 27 GNIPs in China. (line 260) 1. ZHANG Mingjun, WANG Shengjie. A review of precipitation isotope studies in China: Basic pattern and hydrological process[J]. Journal of Geographical Sciences, 2016, 26 (7): 921-938. 2. IAEA/WMO, 2015. Global Network of Isotopes in Precipitation. 2015-11-29.

Q6: Conclusions: This section just summarises the main findings of the project. In

this section explain in more detail how your project helps us to understand processes in these environments more broadly; the paper will have more impact if researchers from elsewhere in the world can see relevance to their studies and a paper in a major international journal such as HESS needs to have broad appeal.

A6: Thanks for your comments. The main objectives of this paper focus on the hydrogeochemical characteristics and transformation relationships in the Weihe River, therefore, the primary results of this research should be listed in the conclusion section. Moreover, we added some sentences in this section to make sure that the research importance is addressed. 'This paper provides systematical methods for hydrogeochemical components analysis which could contribute to the relationship of surface water and groundwater.' 'The research results need to be improved by sufficient local measured data in future research. Moreover, the methods conducted in this paper could offer a new way of research on surface water and groundwater, and the specific results could also provide valuable information for the local water groundwater protection, restoration and management.'

Q7: Figure 1, please show the coordinates of the map. Could you add the groundwater level contours on Fig. 1? Figure 5, is it for showing the results on groundwater samples? not clear. Please be prudent to use the cluster analysis and avoid the false correlation, especially for the groundwater samples. Figure 6,add 'a'and 'b' for the left and right diagrams, respectively. Add the corresponding figure captions. 'Isotopic variation of surface water' There is only one data for one reach of the river. What's the variation?Figure 7, the types of lines are not clear.

A7: Thanks for your comments. (1) Considering that four spots are selected for groundwater levels detection and no coordinates are labeled in the published paper referred, we didn't mark the coordinates in Fig. 1. If reviewers are interested, we could arrange a field trip to this region. (2) We added '(a)' and '(b)' in Figure 5, which represent groundwater variables classification and surface water variables classification respectively. The captain of Fig 5 has been revised. (3) There are five types of lines in Figure

7 whose legends are already shown. The Global meteoric water line and China line co-incide on the top. Then the three lines are Shijiazhuang, Zhengzhou and groundwater samples respectively.

———————————————————

---

## Author Comment (AC3) · 24 Feb 2018

Q1: In my opinion, the pollution in this study area is an aspect that we should pay attention to. But it didn't give any information about the pollutants. Will the pollutants change the hydrochemical contents of the waters?

A1: Thanks for your comments. Please refer to line 75: 'The influence of river pollutants on groundwater is mainly banded and has a relatively small area of influence'. It means that we don't need to take the influence of pollutants for consideration in this study.

[Figure]

Q2: For the STUDY AREA part, the Figure 1 of the sampling sites is not corresponding to the description, and it is not complimentary to enlighten the readers of the study area. For example, the places Xizhangzhuang village of Xiaohe Town and Dongwangqiao village of Liyang Town of the Line 76 are not showing in the figure. It would be helpful, if the contour of water table could be shown in the Figure 1

A2: Thanks for your comments. The scale of the map has restricted showing the places in Fig. 1. Meanwhile, Xizhangzhuang village has occupied almost half area of Xiaohe Town. The monitoring sites are installed in Xizhangzhuang village of Xiaohe Town and Dongwangqiao village of Liyang Town, thus the two villages are considered as substitutes for the regions, which are shown in Fig. 1.

Q3: In the DISCUSSION part, the interpretation of presented hydrochemical ions data could be supported with a more detailed description of the hydrological setting and lithology of the aquifer(s). All the discussion of the hydrochemical contents could go deeper if the geological settings were considered.

A3: This paper mainly focuses on hydrogeochemical characteristics and transformation relationships between surface water and groundwater in the Weihe River. The hydrogeochemical characteristics analysis is basically completed including HCO3-, SO42-, Cl-, Na+, Ca2+ Mg2+, and $\delta D$ and $\delta 18O$ of the surface water.

Q4: Generally, the manuscript is carelessly prepared. Text are readable, however, the abbreviations are not explained, and the figures are not well organized. So it is hard to read this manuscript clearly.

A4: Thanks for your comments. (1) The main text has been revised to be easily understandable, which could be referred to the tracking vision. (2) The abbreviations of 'EC,TDS, RDO, WHO' are cleared in the text, line 97, 116, and 135. (3) The captains of figures are all revised.

Q5: In details, what is the reproductivity of the hydrogen and oxygen isotopes? Please point out all Chinese references (in Chinese) for the international readers that do not understand Chinese language. It is hard to tell the difference of the lines in the Figure 7.

A5:Thanks for your comments. (1) The hydrogen and oxygen isotopes in this study are primarily analyzed as followsïijŽThe reason is that the farmland in the Chaiwan section and the Wangsizhuang section is mainly irrigated using Weihe River water, and the infiltration of irrigation water causes the enrichment of hydrogen and oxygen isotopes in the groundwater. The hydrogen and oxygen isotope characteristics are more similar to those of the Weihe River. The CHW02 and CHW03 sampling points in the Chaiwan section are located in an area affected by river irrigation, and CHW01 is a household well. The hydrogen and oxygen isotope values are CHW03$\geq$CHW02>CHW01. Similarly, WSZ02, WSZ03 and WSZ04 in the Wangsizhuang section are located in an area affected by river irrigation, and WSZ01 is a household well. Thus, the hydrogen and oxygen isotope values are WSZ04$\geq$WSZ03$\geq$WSZ02>WSZ01. (2) Although references in this paper are written by Chinese, the majority are published in top journals by English. Only two of them are in Chinese and have been labeled in the reference section. (3) There are five types of lines in Figure 7 whose legends are already shown. The Global meteoric water line and China line coincide on the top. Then the three lines are Shijiazhuang, Zhengzhou and groundwater samples respectively.

Please also note the supplement to this comment: https://www.hydrol-earth-syst-sci-discuss.net/hess-2017-654/hess-2017-654-AC3-supplement.pdf

**Supplement:**

Q1: In my opinion, the pollution in this study area is an aspect that we should pay attention to. But it didn't give any information about the pollutants. Will the pollutants change the hydrochemical contents of the waters?

A1: Thanks for your comments. Please refer to line 75: 'The influence of river pollutants on groundwater is mainly banded and has a relatively small area of influence'. It means that we don't need to take the influence of pollutants for consideration in this study.

Q2: For the STUDY AREA part, the Figure 1 of the sampling sites is not corresponding to the description, and it is not complimentary to enlighten the readers of the study area. For example, the places Xizhangzhuang village of Xiaohe Town and Dongwangqiao village of Liyang Town of the Line 76 are not showing in the figure. It would be helpful, if the contour of water table could be shown in the Figure 1

A2: Thanks for your comments. The scale of the map has restricted showing the places in Fig. 1. Meanwhile, Xizhangzhuang village has occupied almost half area of Xiaohe Town. The monitoring sites are installed in Xizhangzhuang village of Xiaohe Town and Dongwangqiao village of Liyang Town, thus the two villages are considered as substitutes for the regions, which are shown in Fig. 1.

Q3: In the DISCUSSION part, the interpretation of presented hydrochemical ions data could be supported with a more detailed description of the hydrological setting and lithology of the aquifer(s). All the discussion of the hydrochemical contents could go deeper if the geological settings were considered.

A3: This paper mainly focuses on hydrogeochemical characteristics and transformation relationships between surface water and groundwater in the Weihe River. The hydrogeochemical characteristics analysis is basically completed including $HCO_3^-$, $SO_4^{2-}$, $Cl^-$, $Na^+$, $Ca^{2+}$ $Mg^{2+}$, and $\delta D$ and $\delta^{18}O$ of the surface water.

Q4: Generally, the manuscript is carelessly prepared. Text are readable, however, the abbreviations are not explained, and the figures are not well organized. So it is hard to read this manuscript clearly.

A4: Thanks for your comments.
(1) The main text has been revised to be easily understandable, which could be referred to the tracking vision.
(2) The abbreviations of 'EC,TDS, RDO, WHO' are cleared in the text, line 97, 116, and 135.
(3) The captains of figures are all revised.

Q5: In details, what is the reproductivity of the hydrogen and oxygen isotopes? Please point out all Chinese references (in Chinese) for the international readers that do not understand Chinese language. It is hard to tell the difference of the lines in the Figure 7.

A5:Thanks for your comments.

(1) The hydrogen and oxygen isotopes in this study are primarily analyzed as follows:The reason is that the farmland in the Chaiwan section and the Wangsizhuang section is mainly irrigated using Weihe River water, and the infiltration of irrigation water causes the enrichment of hydrogen and oxygen isotopes in the groundwater. The hydrogen and oxygen isotope characteristics are more similar to those of the Weihe River. The CHW02 and CHW03 sampling points in the Chaiwan section are located in an area affected by river irrigation, and CHW01 is a household well. The hydrogen and oxygen isotope values are CHW03≥CHW02>CHW01. Similarly, WSZ02, WSZ03 and WSZ04 in the Wangsizhuang section are located in an area affected by river irrigation, and WSZ01 is a household well. Thus, the hydrogen and oxygen isotope values are WSZ04≥WSZ03≥WSZ02>WSZ01.

(2) Although references in this paper are written by Chinese, the majority are published in top journals by English. Only two of them are in Chinese and have been labeled in the reference section.

(3) There are five types of lines in Figure 7 whose legends are already shown. The Global meteoric water line and China line coincide on the top. Then the three lines are Shijiazhuang, Zhengzhou and groundwater samples respectively.

**Research on Hydrogeochemical Characteristics and Transformation Relationships between Surface Water and Groundwater in the Weihe River**

*Shibao Lu[1],,Yizi Shang [2,*],Jihong Qu[3] ,Wei Li[1],Zhipeng Gao [4,2], Wujin Li[5, 2], Zhiping Li[2], Furong Yu[2]*

*1 School of Resources and Environment, North China University of Water Resources and Electric Power, Zhengzhou 450045, China;*

*2 State Key Laboratory of Simulation and Regulation of Water Cycle in River Basin, China Institute of Water Resources and Hydropower Research, Beijing 100018, China;*

*3 School of Public Administration, Zhejiang University of Finance and Economics, Hang Zhou 310018, China;*

*4 School of Water Resources and Environment, China University of Geosciences (Beijing), Beijing 100083, China*

*5 Guangdong Hydropower Planning and Design Institute, Guangzhou 510635, China.*

*[*] corresponding author:yzshang@foxmail.com*

**

**

**

**

**

**

**Abstract:** The transforming relationship between surface water and groundwater as well as their origins are the basis for studying the transport of pollutants in river-groundwater systems. Typical section of the river was chosen to sample the surface water and shallow groundwater. Then, a Piper trilinear diagram, Gibbs diagram, ratios of major ions, factor analysis, cluster analysis and other methods were  applied to investigate the hydrogeochemical evolution of surface water and groundwater and determine the formation of hydrogeochemical components in different water bodies. Based on the distribution characteristics of hydrogen and oxygen stable isotopes $\delta D$ and $\delta^{18}O$ and discharge hydrograph separation methods, the relationship between surface water and groundwater in the Weihe River was analyzed. The results  reveal that the river water is a $SO_4$ Cl—Na type and  the groundwater hydrogeochemical types are not the same. The dominant anions are $HCO_3^-$ in the upstream reaches and  $SO_4^{2-}$ and $Cl^-$ in downstream reaches. Hydrogeochemical processes include evaporation and concentration, weathering of rocks, ion exchange, and dissolution infiltration reactions. The $\delta D$ and $\delta^{18}O$ of surface water change little along the river and are more enriched than are those of the groundwater. With the influences of precipitation, irrigation, river recharge and evaporation, the $\delta D$ and $\delta^{18}O$ of shallow groundwater at different sections are not the same. It could be established that surface water recharges groundwater at 5 sections along the Weihe River, and each section has unique recharge intensity and relationship due to its specific hydraulic environment.  Surface water supplies the groundwater, which provides the hydrodynamic conditions for the entry of pollutants into the aquifer.

**Keywords:** hydrogeochemical characteristics; hydrogen and oxygen stable isotopes; surface water-groundwater system; cycle and transformation

**1. Introduction**

The regularity of the water cycle and the conversion between surface water and groundwater is the basis for the study of pollutant transport in river-groundwater systems (Lu et al., 2016a; Shang et al., 2017). Different water bodies have specific  hydrogeochemical characteristics and isotopic signatures due to  different  recharge sources, environments and circulation conditions. Hence, hydrogeochemical characteristics are used for tracking water circulation processes and. Analyses of hydrogeochemical and isotopic characteristics  can effectively reveal the transformation of river water to groundwater (Kanduč et al., 2014). An integration of the hydrogeological and isotopic data could be more reliable and meaningful (Matiatos et al., 2014).

A series of mathematical methods are employed and combined to process the measured data. Descriptive statistics, graphic analysis and multivariate statistical analysis methods are used to determine the hydrogeochemical characteristics (Lu et al., 2015).  Durov diagram (1948), Stiff diagram (1951), Piper diagram (1944) and Gibbs diagram (1970) belong to graphical methods (Rekha et al., 2013). Common multivariate statistical analysis methods include factor analysis (Keesari et al., 2016), principal component analysis (Chattopadhyay and Singh, 2103) and cluster analysis (Zhang et al., 2012). Stable isotopes of $\delta D$ and $\delta^{18}O$ are considered as ideal tracers for tracking various hydrogeochemical processes (Liu et al., 2014). The surface-ground water transformation mechanism is a worldwide topic, about which researches were conducted on many regions via the above methods. Dogramaci et al. (2012) investigated the hydrogeochemical and isotopic characteristics of the Hamersley Basin in northwestern Australia and provided a theoretical basis for the sustainable development of local water resource utilization. A series of methods, such as The descriptive statistical method, the Piper diagram and the main ion component proportion coefficient and the factor analysis method, were employed to study hydrogeochemical characteristics of groundwater in the Sara Wusu aquifer system in the Ordos Basin (Yang et al., 2016). Fuzzy mathematics and multivariate statistical methods were used to study the quality characteristics of surface water and groundwater in the Songnen plain (Zhang et al., 2012). Zeng et al. (2013) investigated the spatial distribution of hydrogeochemical and isotopic characteristics of different water bodies in Tajikistan, including spring water, river water and lake water,  and discussed their origins and environmental significance.  Hydrogen and oxygen stable isotopes ($\delta^{18}O$ and $\delta D$) and electrical conductivity (EC) as typical parameters were adopted to represent the mutual relationship among precipitation, river water and groundwater in Taiwan Douliushan (Peng et al., 2014). Meanwhile, multivariate statistical analysis and isotope analysis methods were used to study the hydraulic linkage between surface water and groundwater and their temporal and spatial variation in the Condamine River in Australia (Martinez et al., 2015). Hydrogen and oxygen isotopes were used to study the relationship of recharge and discharge between the various water bodies on the Portuguese island of Madeira, from which a hydrogeological conceptual model of Madeira Island was established (Prada et al., 2016). By analyzing the hydrogeochemical characteristics of surface water and groundwater in the Heihe River Basin, Nie et al. (2005) identified the transformation relationship between groundwater and surface water in the main stream of Heihe River. Except for the common relationship determination,  a conversion proportion between surface water and groundwater of the Second Songhua River was also calculated quantitatively through the end element method (Zhang et al, 2014). Although many studies related to the chemical and isotopic characteristics of groundwater and surface water have been conducted recently, the relationship between surface water and groundwater transformation is still a prevalent and essential topic in hydrology and water resource studies, hydrogeochemistry, biogeochemistry, and ecohydrology (Wang et al., 2016; Lu et al., 2016b)

The surface water of Weihe River in this paper is seriously polluted and has become a major pollution source for nearby shallow groundwater. This seriously affects the exploitation, utilization and protection of groundwater resources and endangers the ecological safety and the health of the residents (Lu et al., 2016c). This present study has three main objectives as follows: (1) To investigate the hydrogeochemical components formation of surface water and groundwater through the samples taken from several typical sections of the Weihe River Basin; (2) To determine the recharge relationship between surface water and groundwater based on the hydrogeochemical and isotopic characteristics; (3) Tt provide the systematic methods for hydrogeochemical analysis including Piper trilinear diagram, Gibbs diagram, factor analysis and cluster analysis.

**2. Materials and methods**

**2.1 Study area**

The Weihe River Basin with the length of 344.5 km  and  a basin area of 14970 km$^2$, is located in the northern part of the Henan Province, south of the North China Plain. It is the main tributary to the Zhangweinan Canal, which is  tributary to the Haihe River. The Weihe River Basin has a typical warm, temperate, continental monsoon climate. It is cold and dry in the winter hot and rainy in the summer, and the average annual precipitation in the basin is 608 mm (Zhu et al., 2006). The influence of river pollutants on groundwater is mainly banded and has a relatively small range. Considering the shallow groundwater along both sides of the river, a 26.67-km-long segment of the Weihe River between Xizhangzhuang village of Xiaohe Town and Dongwangqiao village of Liyang Town was selected as the study area of approximately 160 km$^2$ (as shown in Fig. 1). The Weihe River Basin is closely related to groundwater, and the polluted river water of the Weihe River is a pollution source of groundwater on both sides of the river. The groundwater is mainly supplied by atmospheric precipitation, lateral seepage, piedmont lateral runoff and canal leakage, and drainage is dominated by artificial extraction and evaporation. Groundwater flows from the southwest to the northeast, which is generally consistent with the topography. The average hydraulic gradient is 1/3000.

[Figure]

**Fig. 1 The location of the study area and the distribution of sampling sections**

2.2 Sample collection

The surface water sampling sites were chosen parallel to the direction of river flow,  while the groundwater sampling sites were  perpendicular to the river flow. There were 5 sections sampled from upstream to downstream between Xiaohe Town and Liyang Town along the Weihe River (i.e., the Zongwan sample section, Hexijie sample section, Chaiwan sample section, Wangsizhuang sample section and Xiwangqiao sample section). A total of 5 surface water samples and 17 groundwater samples were collected in May 2016. Among them, ZWHS, HXJHS, CHWHS, WSZHS and XWQHS were surface water sampling sites, and the others were groundwater sampling sites.  The sampling sites encompassed the band of influence of pollutants from the Weihe River on groundwater. The locations and types of the sampling sites  are shown in Figure 1.

Groundwater was sampled mainly from irrigation wells and drinking wells. Prior to sampling, wells  should be pumped for more than 20 min until the temperature, electrical conductivity (EC), and pH  become stable. Surface water samples were collected from the river bank at a depth of more than 50 cm and kept.  in 500 ml polyethylene bottles, which were  washed with sample water at least three times  before .  After sampling work having been finished, it should be confirmed that no bubbles were left in the bottle, and the outer cap was sealed with sealant to prevent air exchange. Samples were  stored in a refrigerated container at 0 to 4 ℃.

**2.3 Stable hydrogen and oxygen isotopes and hydrogeochemical analysis**

Hydrogen and oxygen isotopes were measured in the laboratory of groundwater science and engineering of
the Ministry of Land and Resources of the Institute of Hydrogeology and Environmental Geology at the Chinese
Academy of Geological Sciences. Analysies were conducted using wavelength scanning-optical cavity ring down
spectroscopy. The ratio of hydrogen and oxygen isotopes (δ) is expressed as the deviation relative to Vienna
VSMOW (Zhao, et al, 2015):

$$\delta(\text{‰}) = \frac{R_{sp} - R_{st}}{R_{st}} \times 1000 \tag{1}$$

where $R_{sp}$ and $R_{st}$ refer to the ratio of D/H (or $^{18}O/^{16}O$) in samples and VSMOW, respectively. When δD and
$\delta^{18}O$ are positive, the samples are enriched with D and $^{18}O$ compared to the VSMOW standard; when they are
negative, the two isotopes are diluted compared to the VSMOW standard (Zhang et al., 2006).

Analysies of water chemicalstry components wasere conducted completed in the laboratory of hydrogeology
at the North China University of Water Resources and Electric Power,. The analyses includinged $Cl^-$, $SO_4^{2-}$, $Na^+$,
$K^+$, $NH_4^+$, $Mg^{2+}$, $HCO_3^-$, $CO_3^{2-}$ and $Ca^{2+}$. Among these ions, $HCO_3^-$ and $CO_3^-$ were detected using acid-base
indicator titration, $Ca^{2+}$ and $Mg^{2+}$ were tested via detected using EDTA titration, and the other ions were through
detected using ion chromatography. pHPH, total dissolved solids (TDS), rugged dissolved oxygen (RDO),
conductivity, redox potential and other indicators were detected *in situ* with a PX.68-smarTROLL MP hand-held
multi-parameter water quality detector.

**2.4 Conversion ratio of surface water to ground watergroundwater**

The stable hydrogen and oxygen isotope method can determine the sources of runoff, the division of river
runoff and the conversion of surface water and groundwater. The principle of division is based on the law of mass
conservation of isotopes (Song et al., 2007), in which the sum of two runoff components is equal to the total flow
of the resultant runoff, and the sum of the tracer flow of the two tracer runoff components is equals to the sum of
the tracer of synthetic runoff (Figure. 2). The calculations basic equations are as follows:

$$Q_t = Q_u + Q_v \tag{2}$$

$$Q_t \cdot C_t = Q_u \cdot C_u + Q_v \cdot C_v \tag{3}$$

$$f = \frac{Q_v}{Q_t} = \frac{C_t - C_u}{C_v - C_u} \tag{4}$$

where $Q$ is the flux, $C$ is the isotope concentrationcomponent, $t$ and $u$ are indicate surface water, and $v$
representsis groundwater. $f$ is the ratio of surface riverwater to ground watergroundwater and is calculated with
δD as a standard.

[Figure]

**Fig. 2 Principle diagram of the discharge hydrograph separation methods**

**3. Results and discussion**

**3.1 Characteristics of main hydrogeochemical components**

The water composition results are shown in Table 1. The groundwater pH of the sampling area was  close to neutral, ranging from 6.83 to 7.81. The TDS values were between 564.66 and 1747.84 mg/L,  all 17 groundwater samples of which exceeded the World Health Organization (WHO) drinking water  threshold of 500 mg/L. As illustrated in Table 1, the average concentrations of groundwater anions was sorted as $HCO_3^->SO_4^{2-}>Cl^-$.     The relationship of the average concentrations of the cations was $Na^+> Ca^{2+}> Mg^{2+}>K^+$. $Na^+$ and $Ca^{2+}$ were dominant, and their concentrations ranged from 74.21 mg/L to 272.00 mg/L and 64 mg/L to 268.80 mg/L, respectively, with average values of 182.78 mg/L and 121.69 mg/L.

The pH of the Weihe River in the study area ranged from 8.03 to 8.22, which  was  weakly alkaline. The TDS with an average value of 1473.74 is generally higher than that of groundwater. The sorting results of  the concentrations of anions and cations in surface water were $SO_4^{2-}> Cl^-> HCO_3^-$ and $Na^+> Ca^{2+}> Mg^{2+}>K^+$, respectively. The concentration of $SO_4^{2-}$ in the river water ranged from 627.07 mg/L to 664.06 mg/L, with an average value of 647.12 mg/L. The concentration of $Cl^-$ ranged from 325.95 mg/L to 391.57 mg/L, with an average concentration of 365.89 mg/L. The concentrations of $Na^+$ and $Ca^{2+}$ ranged from 294.47 mg/L to 314.27 mg/L and 94.40 mg/L to 115.20 mg/L, respectively, and their mean values were 305.58 mg/L and 107.52 mg/L. As seen in Table 1, there is no significant change in the ion concentration between the upstream and downstream parts of the Weihe River.

According to WHO  drinking water standards, except for $K^+$ and pH, the other measured components of surface water and groundwater all exceeded the maximum acceptable values in the study area.  Under such conditions, both surface water and groundwater along the Weihe River could not be considered as  suitable drinking water sources.

**Table 1 Analytical results of water quality in the study area**

| Ion content/(mg.L$^{-1}$) | | pH | TDS | Na$^+$ | K$^+$ | Mg$^{2+}$ | Ca$^{2+}$ | Cl$^-$ | SO$_4^{2-}$ | HCO$_3^-$ |
|---|---|---|---|---|---|---|---|---|---|---|
| Groundwater (17) | Minimum | 6.73 | 564.66 | 74.21 | 6.20 | 57.76 | 64.00 | 102.17 | 116.11 | 461.75 |
| | Maximum | 7.81 | 1747.84 | 272.00 | 34.26 | 162.38 | 268.80 | 640.13 | 833.33 | 735.15 |
| | Average | 7.30 | 1170.00 | 182.78 | 16.58 | 110.23 | 121.69 | 275.24 | 307.52 | 635.88 |

| | | | | | | | | | |
|---|---|---|---|---|---|---|---|---|---|
| Surface water (5) | Minimum | 8.03 | 1401.32 | 294.47 | 24.23 | 49.90 | 94.40 | 325.95 | 627.07 | 282.52 |
| | Maximum | 8.22 | 1518.71 | 314.27 | 28.35 | 59.54 | 115.20 | 391.57 | 664.06 | 385.80 |
| | Average | 8.11 | 1473.74 | 305.58 | 26.40 | 53.79 | 107.52 | 365.89 | 647.12 | 350.56 |
| WHO drinking water standards | | 6.5~8.5 | 500 | 200 | 100 | 30 | 75 | 200 | 200 | 200 |
| Over-standard rate of groundwater (%) | | 0 | 100 | 47 | 0 | 100 | 70 | 70 | 65 | 100 |
| Over-standard rate of surface water (%) | | 0 | 100 | 100 | 0 | 100 | 100 | 100 | 100 | 100 |

**3.2 Hydrogeochemical characteristics**

The Piper diagram is widely applied as graphical method for settling  hydro-geological problems. According to the analytical results, the Piper diagram of the hydrogeochemical composition of all water samples in the study area is shown in Fig. 3. The results illustrate that the chemical type of surface water  is  $SO_4$ Cl—Na , indicating that the surface water is uniform across the study region. From upstream to downstream along the Weihe River, the water  chemical type of each groundwater section is as follows: The Zong Wan section and Hexijie section are mainly $HCO_3$--Mg Na types, the Chaiwan section is mainly the $SO_4$ Cl—Mg Na type, the Wangsizhuang section is mainly the $HCO_3$ $SO_4$ Cl--Mg Na type, and the Xiwangqiao section is mainly the $HCO_3$ Cl—Mg Ca Na type. From the recharge area to the discharge area, the chemical types of groundwater usually  change in the following ways: $HCO_3^-$ —$SO_4^{2-}$ —$Cl^-$. Based on the results of groundwater chemical types , it could be concluded that $HCO_3^-$ is dominant in groundwater on both sides of river in the upstream section, whereas $SO_4^{2-}$ and $Cl^-$ are dominant in the middle and lower reaches. The water chemical types can indirectly verify that the groundwater flow on both sides of river.

[Figure]

**Fig. 3 Piper diagram of water chemistry for surface and groundwater in the study area**

**3.3 Analysis of formation function of water chemical composition**

**3.3.1 Analysis of formation function based on the Gibbs diagram**

The Gibbs diagram can clearly indicate whether the chemical components of river and groundwater are the precipitation dominance type, rock dominance type or evaporation crystallization dominance type. It is an efficient  way to qualitatively determine the effects of regional rocks, atmospheric precipitation, and evaporation concentration on river water components (Wang et al., 2010). Generally, samples with low TDS and high $Na^+/(Na^++Ca^{2+})$ or $Cl^-/(Cl^-+HCO_3^-)$ ratios (close to 1),  mainly distributed in the lower-right corner,  indicate precipitation dominance. Samples with slightly high TDS and $Na^+/(Na^++Ca^{2+})$ or $Cl^-/(Cl^-+HCO_3^-)$ ratios of approximately 0.5 or less than 0.5 . mainly distributed in the middle zone,  indicate rock dominance. Samples with very high TDS and large $Na^+/(Na^++Ca^{2+})$ or $Cl^-/(Cl^-+HCO_3^-)$ ratios,  mainly distributed in the upper-right corner, indicat evaporation crystallization dominance type, reflecting the influence of evaporation on arid areas (Sun et al., 2014).

The ion concentrations of the 5 river surface water samples and 17 groundwater samples from the study area are shown on a Gibbs diagram in Fig. 4. Apparently,  the surface water samples  are distributed the upper-right corner of the diagram with a $Na^+/(Na^++Ca^{2+})$ or $Cl^-/(Cl^-+HCO_3^-)$ ratio greater than 0.5 and  a high content of TDS, which means that surface water has an evaporation crystallization dominance origin. The groundwater samples differ in the two criterions  are mainly distributed in the evaporation crystallization dominance region, slightly toward to the rock dominance region, indicating that the chemical composition of water is both controlled by evaporation crystallization and rock weathering.

[Figure]

**Fig. 4 Gibbs plots of the  river and groundwater chemistry in the study area, confirming the type of evaporation-crystallization dominance, rock weathering dominance and precipitation dominance.**

**3.3.3 Analysis of formation function based on multivariate statistics**

To further analyze the hydrogeochemical formation functions, factor analysis and R cluster analysis were conducted on surface and ground water samples using TDS, Cl⁻, SO₄²⁻, HCO₃⁻, Na⁺, K⁺, Ca²⁺, and

Mg²⁺ as the original parameters.

The initial KMO value of the groundwater sample was 0.596. According to the KMO test standard,  the original variable is barely suitable for factor analysis when

0.5<KMO<0.6. According to the  results shown in (Table 2, the formation function of the groundwater chemical composition could be summarized in 3 factors, and the cumulative contribution rate of  all 3

factors is 75%. The Ca²⁺, Cl⁻, SO₄²⁻ and TDS of factor 1 have higher positive load and the coefficient of Ca²⁺ and

SO₄²⁻ is large, indicating that there may be weathering of calcium feldspar, dissolution of gypsum, or oxidation of pyrite. A large coefficient of Cl⁻ suggests that there may be leaching of halite. The Na⁺, Mg²⁺ and HCO₃⁻

of factor 2 have higher positive loads, which indicates possible weathering of carbonate or silicate. The alternating adsorption of cations between Na⁺ and Ca²⁺ causes the content of Na⁺ to increase. The K⁺ content of factor 3 is large, which indicates  weathering of feldspar. Meanwhile, surface water samples can be summarized in

2 factors with the contribution rate of 91%. The K⁺, HCO₃⁻ and TDS of factor 1 have higher positive loads, indicating the possible weathering of carbonate and feldspar. The Mg²⁺, Na⁺ and Cl⁻ of factor 2

have  higher positive load, which indicates the possible dissolution of halite and the alternate adsorption of cations between Na⁺ and Ca²⁺ causes the content of Na⁺ to increase.

Cluster analysis can be simplified as the identification of the relationship between large-scale samples. R

cluster analysis is used to classify variables, while Q cluster analysis is used to classify samples (Sun and Gui,

2013). The results of the R cluster analysis are shown in Fig. 5. The groundwater components can be divided into 3 groups, which is totally consistent with the results of the factor analysis.

Surface water ions can be divided into 2 groups which differ from the results of factor analysis. The first group includes

TDS, Cl⁻, HCO₃⁻, Na⁺, K⁺ and Mg²⁺, and the second group includes Ca²⁺ and SO₄²⁻. This indicates the presence of gypsum dissolution.

In summary, a variety of complex hydrogeochemical processes may have occurred in the study area, such as concentration through evaporation, rock weathering, cation alternate adsorption, oxidation and dissolution.

**Table 2 Factor analysis composition coefficient of ground and surface water**

| Parameter variable | Groundwater | | | Surface water | |
|---|---|---|---|---|---|
| | Factor 1 | Factor 2 | Factor 3 | Factor 1 | Factor 2 |
| Ca²⁺ | 0.889 | -0.083 | 0.339 | 0.206 | -0.970 |
| Na⁺ | 0.261 | 0.799 | -0.041 | 0.650 | 0.738 |
| K⁺ | 0.061 | 0.081 | 0.916 | 0.783 | 0.540 |
| Mg²⁺ | 0.298 | 0.702 | 0.265 | 0.106 | 0.986 |
| Cl⁻ | 0.735 | 0.259 | 0.323 | 0.479 | 0.817 |
| SO₄²⁻ | 0.760 | 0.299 | -0.046 | -0.883 | 0.351 |
| HCO₃⁻ | -0.004 | 0.915 | 0.010 | 0.812 | 0.151 |
| TDS | 0.762 | 0.197 | -0.208 | 0.972 | 0.200 |
| Characteristic value | 3.472 | 1.492 | 1.037 | 4.856 | 2.446 |
| Contribution rate% | 43.394 | 18.649 | 12.965 | 60.704 | 30.578 |
| Cumulative contribution rate | 43.394 | 62.043 | 75.008 | 60.704 | 91.282 |

[Figure]

**Fig. 5 R type cluster analysis. (a) groundwater variables; (b) surface water variables.**

**3.4 Isotopic characteristics and transformation relationships of surface water and groundwater**

3.4.1 Isotopic variation characteristics

δD and $\delta^{18}$O of surface water and groundwater and the *d* value of deuterium excess in the study area are listed in Table 3, where *d*=δD-8$\delta^{18}$O.

As seen in Table 3, δD and $\delta^{18}$O of surface water are more enriched than those of groundwater. The variation  in δD and $\delta^{18}$O of surface water  is relatively small. $\delta^{18}$O ranges from -7.8‰

to -7.6‰ with an average value of -7.7‰, and δD ranges from -59‰ to -57‰ with an average value of -58‰. The range of $\delta^{18}$O for shallow groundwater is from -9.4‰ to -7.7‰, with an average value of -8.55‰; the range of δD

is from -59‰ to -69‰, with an average value of -63.3‰. The *d* value of deuterium excess  is positive and less than 10 of the atmospheric precipitation intercept. The value of surface water is less than that of shallow groundwater, indicating that the recharge sources of surface water and groundwater are subject to evaporation effects but  shallow groundwater is less influenced .

Generally, the isotopic  concentrations of river water bodies increase from upstream to downstream because of the isotopic fractionation  caused by the evaporation of water. The fractionation effect on the isotopes could be greater as the location is closer to the lower reaches of the river (Liu et al., 2014).

As shown in Fig. 6, it is apparent that δD and $\delta^{18}$O become more enriched as the river flows downstream, in which $\delta^{18}$O declines in HXJHS, probably because there is a lake in the vicinity of the upstream reaches and river water supplies the lake.

For groundwater, the values of δD and $\delta^{18}$O in the Zongwan section, Hexijie section and Xiwangqiao section become more depleted as the distance between sampling points and Weihe River increases. As the sample location is closer to the river,  the δD and $\delta^{18}$O values are more similar to the surface water, revealing that the influence of surface water on groundwater decreases with increasing distance. In contrast, the influence of precipitation and irrigation infiltration recharge on groundwater is enhanced. The values of δD

and $\delta^{18}$O for the Chaiwan section and the Wangsizhuang section become enriched as the distance between the sampling points and the Weihe River increases. The reason is probably that  the farmland in the Chaiwan section and the Wangsizhuang section is mainly irrigated using Weihe River water, and the infiltration of irrigation water causes the enrichment of hydrogen and oxygen isotopes in the groundwater. The hydrogen and oxygen isotope characteristics are more similar to those of the Weihe River. The CHW02 and CHW03 sampling points in the Chaiwan section are located in an area affected by river irrigation, and CHW01 is a household well. The hydrogen and oxygen isotope values are CHW03≥CHW02>CHW01. Similarly, WSZ02, WSZ03 and WSZ04 in the Wangsizhuang section are located in an area affected by river irrigation, and WSZ01 is a household well. Thus, the hydrogen and oxygen isotope values are WSZ04≥WSZ03≥WSZ02> WSZ01.

**Table 3 δD, δ$^{18}$O and $d$ values of water samples in study area**

| Water sample type | No. | δ D/‰ | δ$^{18}$O/‰ | $d$/‰ | $f$/% |
|---|---|---|---|---|---|
| Surface water | ZWHS | -59 | -7.7 | 2.6 | |
| | HXJHS | -58 | -7.8 | 4.4 | |
| | CHWHS | -58 | -7.7 | 3.6 | |
| | WSZHS | -58 | -7.7 | 3.6 | |
| | XWQHS | -57 | -7.6 | 3.8 | |
| | average | -58 | -7.7 | 3.6 | |
| Shallow groundwater | ZW01 | -62 | -8.6 | 6.8 | 33.3 |
| | ZW02 | -63 | -8.6 | 5.8 | 25 |
| | ZW03 | -63 | -8.6 | 5.8 | 25 |
| | ZW04 | -69 | -9.4 | 6.2 | 10 |
| | D01 | -65 | -8.7 | 4.6 | 14.3 |
| | D02 | -66 | -8.9 | 5.2 | 12.5 |
| | D03 | -60 | -8 | 4 | 50 |
| | CHW01 | -63 | -8.6 | 5.8 | 20 |
| | CHW02 | -62 | -8.4 | 5.2 | 25 |
| | CHW03 | -62 | -8.3 | 4.4 | 25 |
| | WSZ01 | -68 | -9.2 | 5.6 | 10 |
| | WSZ02 | -63 | -8.6 | 5.8 | 20 |
| | WSZ03 | -63 | -8.5 | 5 | 20 |
| | WSZ04 | -63 | -8.5 | 5 | 20 |
| | D11 | -59 | -7.7 | 2.6 | 50 |
| | D12 | -62 | -8.4 | 5.2 | 20 |
| | D13 | -63 | -8.4 | 4.2 | 16.7 |
| | Average | -63.3 | -8.55 | 5.1 | |

[Figure]

**Fig. 6 Isotopic variation of surface water. (a) δ18O; (b) δD**

[Figure]

**Fig. 7 Relationship between δD and δ18O of surface water and groundwater**

According to the 27 Global Network of Isotopes in Precipitations (GNIPs) set up in China by the International Atomic Energy Association (IAEA), the monitoring sites that could be considered as substitutes for the study area are at Shijiazhuang and Zhengzhou (IAEA/WMO, 2015; Zhang and Wang, 2016). The meteoric water line of $\delta D=6.75\ \delta^{18}O-5.12$ in Zhengzhou is close to the characteristic line for hydrogen and oxygen isotopes of samples in  this study  region (Fig. 7), which is taken as the local meteoric water line (LMWL). When the compositions of δD and $\delta^{18}O$ are compared to the meteoric water line, the source of the local river water and shallow groundwater and their mutual transformation relationship can be distinguished. From the trend line  of the groundwater sample points, the relationship between δD and $\delta^{18}O$ is $\delta D=6.1873\delta^{18}O-10.375$ with a correlation coefficient of 0.9285. From  the trend line  of the  groundwater and surface water sample points, the relationship between δD and $\delta^{18}O$ is fitted with $\delta D=6.19328\delta^{18}O-10.321$ and the correlation coefficient is 0.9585. The two trend lines are sameextremely close with , and the high related coefficient is very high. The surface water sample points are located in the direction of the groundwater trend line that extends to the right. δD and δ¹⁸O are relatively enriched, indicating that the sources of surface water and groundwater are the same and that there is a hydraulic connection. The hydraulic connection between the two is, i.e. a single-line infiltration of the surface river water into groundwater. The trend line is close to the local meteoric water line (LMWL) and the slope is small, which meansindicating that surface water and groundwater are recharged from meteoric water but are also subject to the evaporation, resulting in the enrichment of hydrogen and oxygen isotopes.

3.4.2 Estimation of recharge capacity of river water to groundwater

According to the stable isotope signatures of the water samples, the calculated results of ratio *f* of groundwater recharge to river water are shown in Table 43. As illustrated in Fig. 8, except for regular pattern, tThe ratio *f* of surface water infiltration to recharge groundwater in each observation section also shows has a differencet, but regular, pattern, and the results are shown in Figure 8.

In the Zongwan section (Fig.gure 8a), as the distance between groundwater sampling sites and the river increased (ZW01 toward ZW04), the ratio of surface water infiltration to groundwater recharge (*f*) decreased from 33.3% to 10%, confirming indicating that the river water recharges groundwater in this section and the direction of groundwater flow is from ZW01 toward ZW04. The infiltration rates at D01 and D02 in the Hexijie section (Figure Fig. 8b) are 14.3% and 12.5% , respectively, andwith from thea decreasing trend, it can be deducedindicating that there is a small amount of river water recharging groundwater in that section, with a direction of groundwater flow from D01 toward D02. The ratio of surface water infiltration to groundwater at D03 is as high as 50%, indicating which means that the river mainly recharges the artificial lake that exists near D03 in the Hexijie section. The ratio of surface water infiltration to groundwater in the Chaiwan section (Fig.ure 8C) increases from 20% to 25% as the distance increases between groundwater sampling sites and the river, whereas it increases from 10% to 20% in the Wangsizhuang section (Figure Fig. 8D). This may be associated with the unique river trend of the two sections. The Chaiwan section and the Wangsizhuang section are located near the right corner of the river, where the influence of the river water on groundwater is complicated, but the river is the main supply ier of groundwater. The groundwater flow line is anthe enclosed space where therewater is no waternot exchanged with the outside world. ISome input values like δD and δ¹⁸O remain constant along the entire streamline, such as δD and δ¹⁸O, inferring. Therefore, it is possible to interpret that WSZ01, WSZ02, and WSZ03 are on the same streamline. At the same time, because farmland is primarily irrigated by water from the Weihe River, irrigation water infiltrates the soil to recharge groundwater, resulting in the enrichment of hydrogen and oxygen isotopes. For the Xiwangqiao section (Fig.ure 8e), the ratio of river water infiltration to groundwater at D11 is close to 50%, whereas it is 20% and 16.7% at D12 and D13, respectively. This is primarily because D11 is located in the convexity of the river, where it is significantly eroded with a large amount of infiltration.

[Figure]

                         (a) Zongwan section                            (b) Hexijie section

                    (c) Chaiwan section                        (d) Wangsizhuang section

                                (e) Xiwangqiao section

**Fig. 8 Recharge directions between surface water and groundwater at all sampling sections**

**358 4. Conclusions**

This paper provides systematical methods for hydrogeochemical components analysis which could contribute to the relationship of surface water and groundwater. The main results are concluded as follows:

(1) The surface water components of Weihe River  have no significant spatial variations, but the ion concentrations of groundwater samples from 5 sections are different. The cation concentrations of surface water and groundwater are consistent, with $Na^+ > Ca^{2+} > Mg^{2+} > K^+$. The relative concentrations of anions in groundwater are $HCO_3^- > SO_4^{2-} > Cl^- > NO_3^-$, whereas the relative concentrations of anions in the surface water are $SO_4^{2-} > Cl^- >$

$HCO_3^->NO_3^-$. The surface water in all sections of the Weihe River is the $SO_4$ Cl—Na type, whereas the hydrogeochemical types of groundwater are not the same different. $HCO_3^-$ dominates in the groundwater in the upper reaches of the river, and while $SO_4^{2-}$ and $Cl^-$ dominate in the middle and lower reaches.

(2) By usingBased on a Gibbs diagram, factor analysis and cluster analysis, we established that the geochemical processes of the Weihe River Basin include concentration by evaporation, rock weathering, cation alternate adsorption and dissolution. Because surface water is an open system, the source of ions in a water body is greatly influenced by human activity and atmospheric precipitation, whereas the factors contributing to the formation of water chemistry are more complex.

(3) The isotope results show that δD and $δ^{18}O$ of the surface water in the Weihe River of small variations varies little areand is more enriched than those of the groundwater is. Affected by rainfall, irrigation, river recharge and evaporation, tThe shallow groundwater at different sections is affected by rainfall, irrigation, river recharge and evaporation, resulting inhave different δD and $δ^{18}O$ values. By analyzing hydrogen and oxygen isotopic characteristics of surface water and groundwater in different sections and using the segmentation of flow duration curve, it was established that surface water recharges groundwater at 5 sections along the Weihe River, and each section has unique recharge intensity and relationship due to its specific unique hydraulic environment.

However, dDue to the lack of local isotope monitoring data for meteoric water, the Zhengzhou meteoric water line was used to analyze the isotopic characteristics of surface water and groundwater. The existing household wells, were used as groundwater sampling points, . Because they are affected by towns and villages surrounding the Weihe River, causing that groundwater sampling points cannot be fully symmetric and isometric relative to the Weihe River. As such, tThe research results need to be improved by sufficient monitoring more local measured complete data in future research. Moreover, the methods conducted in this paper could offer efficient ways of research on surface water and groundwater, and the specific results could also provide valuable information for the local water groundwater protection, restoration and management.

**Acknowledgments**

The study was financially supported by Non-Profit Industry Specific Research Projects of Ministry of Water Resources, China, Grant NO: 201401041and 201501008, the Open Research Fund of State Key Laboratory of Simulation and Regulation of Water Cycle in River Basin (China Institute of Water Resources and Hydropower Research), Grant NO: IWHR-SKL-201208, and Science and Technology Research Key Project of the Education Department of Henan Province, Grant NO: 14A170006.

**References**

Chattopadhyay Pallavi Banerjee, Singh, V. S. Hydrochemical evidences: Vulnerability of atoll aquifers in Western Indian Ocean to climate change[J]. *Global & Planetary Change*, 2013, 106:123-140.

Dogramaci Shawan, Skrzypek Grzegorz, Dodson Wade Pauline F. Stable isotope and hydrochemical evolution of groundwater in the semi-arid Hamersley Basin of subtropical northwest Australia[J]. *Journal of Hydrology*, 2012, 475 (26): 281-293.

IAEA/WMO, 2015. Global Network of Isotopes in Precipitation. 2015-11-29.

Kanduč T, Grassa F, McIntosh J, et al. A geochemical and stable isotope investigation of groundwater/surface-water interactions in the Velenje Basin, Slovenia[J]. Hydrogeology Journal, 2014, 22(4): 971-984.

Keesari Tirumalesh, Ramakumar K.L., Chidambaram S., Pethperumal S., Thilagavathi R. Understanding the hydrochemical behavior of groundwater and its suitability for drinking and agricultural purposes in Pondicherry area, South India – A step towards sustainable development[J]. *Groundwater for Sustainable Development*, 2016, 2-3, 143-153.

Liu Fen, Wang Shui-Xian, Lan Yong-Chao, Hu Xing-lin. Environmental isotopes and exchanges of surface–water-groundwater system in the Zhangye basin of Heihe River watershed[J]. *South-to-North Water Transfers and Water Science& Technology*, 2014, 12(2): 92-96. (In Chinese)

Lu Shi-Bao, Bao Hai-Jun, Pan Hu-Lin. Urban water security evaluation based on similarity measure model of Vague sets[J]. *International Journal of Hydrogen Energy*, 2016c, 41(35):15944-15950.

Lu Shi-Bao, Wang Jian-Hua, Pei Liang. Study on the Effects of Irrigation with Reclaimed Water on the Content and Distribution of Heavy Metals in Soil[J]. *International Journal of Environmental Research & Public Health*, 2016b, 13(3): 298.

Lu Shi-Bao, Pei Liang, Bai Xiao. Study on method of domestic wastewater treatment through new-type multi-layer artificial wetland[J]. *International Journal of Hydrogen Energy*, 2015, 40(34):11207-11214.

Lu Shi-Bao, Zhang Xiao-Ling, Bao Hai-Jun, Skitmore Martin. Review of social water cycle research in a changing environment[J]. *Renewable & Sustainable Energy Reviews*, 2016a, 63:132-140.

Matiatos I, Alexopoulos A, Godelitsas A. Multivariate statistical analysis of the hydrogeochemical and isotopic composition of the groundwater resources in northeastern Peloponnesus (Greece)[J]. Science of the Total Environment, 2014, 476: 577-590.

Martinez Jorge L., Raiber Matthias, Cox Malcolm E. Assessment of groundwater–surface water interaction using long-term hydrochemical data and isotope hydrology: Headwaters of the Condamine River, Southeast Queensland, Australia[J]. *Science of the Total Environment*, 2015, 536:499-516.

Ma Rui, Dong Qi-Ming, Sun Zi-Yong, Zheng Chun-Miao. Using heat to trace and model the surface water-groundwater interactions: a review[J]. *Geological Science and Technology Information*, 2013, 32 (2):131-137.

Nie Zhen-Long, Chen Zong-Yu, Chen Xu-Xue, Hao Ming-Lin, Zhang Guang-Hui. The chemical information of the interaction of unconfined groundwater and surface water along the Heihe River, Northwestern China[J]. *Journal of Jilin University (Earth Science Edition)*, 2005, 35(1):48-51.

Prada Susana, Cruz J. Virgílio, Figueira Celso. Using stable isotopes to characterize groundwater recharge sources in the volcanic island of Madeira, Portugal[J]. *Journal of Hydrology*, 2016, 536:409-425.

Peng Tsung-Ren, Lu Wan-Chung , Chen Kuan-Yu, Zhan Wen-Jun, Liu Tsung-Kwei. Groundwater-recharge connectivity between a hills-and-plains' area of western Taiwan using water isotopes and electrical conductivity[J]. *Journal of Hydrology*, 2014, 517:226-235.

Rekha V B, George A V, Rita M. A Comparative Study of Ion Chemistry of Groundwater Samples of Typical Highland and Midland Sub-watersheds of the Manimala River Basin, Kerala, South India[J]. Environmental Research, Engineering and Management, 2013, 66(4): 22-33.

Shang Yi-Zi, Lu Shi-Bao, Li Xiao-Fei, Hei Peng-Fei, Lei Xiao-Hui, Gong Jia-Guo, Liu Jia-Hong, Zhai Jia-Qi, Wang Hao. Balancing development of major coal bases with available water resources in China through 2020[J]. *Applied Energy*, 2017, 194:735-750.

Song Xian-Fang, Liu Xiang-Chao, Xia Jun, Yu Jing-Jie, Tang Chang-Yuan. A study of interaction between surface water and groundwater using environmental isotopes in Huaisha River basin[J]. *Science in China (Series D)*, 2007, 37(1):102-110.

Sun Lin-Hua, Gui He-Rong. Statistical analysis of deep groundwater geochemistry from Taoyuan Coal Mine, northern Anhui Province[J]. *Journal of China Coal Society*, 2013, 38(s2):442-447.

Sun Peng-Fei, Yi Ji-Tao, Xu, Guang-Quan. Characteristics of water chemistry and their influencing factors in subsidence waters in the Huainan and Huaibei mining areas, Anhui Province[J]. *Journal of China Coal Society*, 2014, 39(7):1345-1353.

Wang Jian-Hua, Lu Shi-Bao, Pei Liang. Study on rules of dynamic variation of nitrogen in soil after reclaimed water drip irrigation[J]. *International Journal of Hydrogen Energy*, 2016, 41(35):15938-15943.

Wang Ya-Ping, Wang Lan, Xu Chun-Xue, Yang Zhong-Fang, Ji Jun-Feng, Xia Xue-Qi, An Zi-Yi, Yuan Jian. Hydro-geochemistry and genesis of major ions in the Yangtze River, China[J]. *Geological Bulletin of China*, 2010, 29(2-3):446-456. (In Chinese)

Yang Qing-Chun, Wang Lu-Chen, Ma Hong-Yun, Yu Kun, Martín Jordi Delgado. Hydrochemical characterization and pollution sources identification of groundwater in Salawusu aquifer system of Ordos Basin, China[J]. *Environmental Pollution*, 2016,

216:340-349.

Zhao Shi-Kun, Pang Shuo-Guang, Wen Rong, Liu Zhong-Fang. Influence of below-cloud secondary evaporation on stable isotope composition in precipitation in the Haihe River Basin, China[J]. *Progress in Geography*, 2015, 34 (8):1031-1038.

Zeng Hai-Ao, Wu Jing-Lu. Water isotopic and hydrochemical characteristics and causality in Tajikistan[J]. *Advances in Water Science*, 2013, 24(2):272-279.

Zhang Bing, Song Xian-Fang, Zhang Ying-Hua, Han Dong-Mei, Tang Chang-Yuan, Yu Li-Lei, Ma Ying. Hydrochemical characteristics and water quality assessment of surface water and groundwater in Songnen plain, Northeast China[J]. *Water Research*, 2012, 46(8):2737-2748.

Zhang Bing, Song Xian-Fang, Zhang Ying-Hua, Han Dong-Mei, Yang Li-Hu, Tang Chang-Yuan. Relationship between surface water and groundwater in the second Songhua River basin[J]. *Advances in Water Science*, 2014, 25(3):336-347.

Zhang Mingjun, Wang Shengjie. A review of precipitation isotope studies in China: Basic pattern and hydrological process[J]. *Journal of Geographical Sciences*, 2016, 26 (7): 921-938.

Zhu Xin-Jun, Wang Zhong-Gen , Li Jian-Xin, Yu Lei, Wang Jin-Gui. Applications of SWAT model in Zhang Wei River Basin[J]. *Progress in Geography*, 2006, 25(5):106-111.

Zhang Ying-Hua, Wu Yan-Qing, Wen Xiao-Hu, Su Jian-Ping. Application of environmental isotopes in water cycle[J]. *Advances in Water Science*, 2006, 17(5):738-747.